# Antidiabetic and Immunoregulatory Activities of Extract of *Phyllanthus emblica* L. in NOD with Spontaneous and Cyclophosphamide-Accelerated Diabetic Mice

**DOI:** 10.3390/ijms24129922

**Published:** 2023-06-08

**Authors:** Cheng-Hsiu Lin, Yueh-Hsiung Kuo, Chun-Ching Shih

**Affiliations:** 1Department of Internal Medicine, Fengyuan Hospital, Ministry of Health and Welfare, Taichung City 42055, Taiwan; kenny0139@gmail.com; 2Department of Chinese Pharmaceutical Sciences and Chinese Medicine Resources, China Medical University, Taichung City 40402, Taiwan; kuoyh@mail.cmu.edu.tw; 3Department of Nursing, College of Nursing, Central Taiwan University of Science and Technology, Taichung City 40601, Taiwan

**Keywords:** diabetes, immunoregulatory, *Phyllanthus emblica* L., non-obese diabetes, cyclophosphamide, interleukin 17 (IL-17)

## Abstract

Oil-Gan, also known as emblica, is the fruit of the genus *Phyllanthus emblica* L. The fruits are high in nutrients and display excellent health care functions and development values. The primary aim of this study was to investigate the activities of ethyl acetate extract from *Phyllanthus emblica* L. (EPE) on type 1 diabetes mellitus (T1D) and immunoregulatory activities in non-obese diabetes (NOD) mice with spontaneous and cyclophosphamide (Cyp)-accelerated diabetes. EPE was vehicle-administered to spontaneous NOD (S-NOD) mice or Cyp-accelerated NOD (Cyp-NOD) mice once daily at a dose of 400 mg/kg body weight for 15 or 4 weeks, respectively. At the end, blood samples were collected for biological analyses, organ tissues were dissected for analyses of histology and immunofluorescence (IF) staining (including expressions of Bcl and Bax), the expression levels of targeted genes by Western blotting and forkhead box P3 (Foxp3), and helper T lymphocyte 1 (Th1)/Th2/Th17/Treg regulatory T cell (Treg) cell distribution by flow cytometry. Our results showed that EPE-treated NOD mice or Cyp-accelerated NOD mice display a decrease in levels of blood glucose and HbA1c, but an increase in blood insulin levels. EPE treatment decreased blood levels of IFN-γ and tumor necrosis α (TNF-α) by Th1 cells, and reduced interleukin (IL)-1β and IL-6 by Th17 cells, but increased IL-4, IL-10, and transforming growth factor-β1 (TGF-β1) by Th2 cells in both of the two mice models by enzyme-linked immunosorbent assay (ELISA) analysis. Flow cytometric data showed that EPE-treated Cyp-NOD mice had decreased the CD4^+^ subsets T cell distribution of CD4^+^IL-17 and CD4^+^ interferon gamma (IFN-γ), but increased the CD4^+^ subsets T cell distribution of CD4^+^IL-4 and CD4^+^Foxp3. Furthermore, EPE-treated Cyp-NOD mice had decreased the percentage per 10,000 cells of CD4^+^IL-17 and CD4^+^IFNγ, and increased CD4^+^IL-4 and CD4^+^Foxp3 compared with the Cyp-NOD Con group (*p* < 0.001, *p* < 0.05, *p* < 0.05, and *p* < 0.05, respectively). For target gene expression levels in the pancreas, EPE-treated mice had reduced expression levels of inflammatory cytokines, including IFN-γ and TNF-α by Th1 cells, but increased expression levels of IL-4, IL-10, and TGF-1β by Th2 cells in both two mice models. Histological examination of the pancreas revealed that EPE-treated mice had not only increased pancreatic insulin-expressing β cells (brown), and but also enhanced the percentage of Bcl-2 (green)/Bax (red) by IF staining analyses of islets compared with the S-NOD Con and the Cyp-NOD Con mice, implying that EPE displayed the protective effects of pancreas β cells. EPE-treated mice showed an increase in the average immunoreactive system (IRS) score on insulin within the pancreas, and an enhancement in the numbers of the pancreatic islets. EPE displayed an improvement in the pancreas IRS scores and a decrease in proinflammatory cytokines. Moreover, EPE exerted blood-glucose-lowering effects by regulating IL-17 expressions. Collectively, these results implied that EPE inhibits the development of autoimmune diabetes by regulating cytokine expression. Our results demonstrated that EPE has a therapeutic potential in the preventive effects of T1D and immunoregulation as a supplementary.

## 1. Introduction

Type 1 diabetes mellitus (T1D) is an autoimmune disease that arises from the selective and progressive loss of insulin-producing β cells by means of self-reactive T lymphocytes [1,2]. The pathogenesis of T1D is complex which is associated with the activation of antigen-specific helper T lymphocyte 1 (Th1) cells. Proinflammatory cytokines play a major role in autoimmune diabetes [3,4].

Previous evidence has shown that models of T cell differentiation focused on the dichotomy between Th1 and helper T lymphocyte 2 (Th2) responses, with type 1 diabetes (T1D) being observed primarily as a Th1-mediated pathology [5]. The novel of Th17 cells [6,7] indicated a primary revision hypothesis of the Th1/Th2 prototype [8], and there is a distinct possibility that tissue-specific autoimmunity could be determined by interleukin (IL)-17-producing T cells instead of Th1 cells.

Recent animal studies suggest that the Th17 subset is necessary for the early stages of diabetes development [9,10], but IL-17 displayed an up-regulation not essentially following this cytokine [5]. IL-1β plays a role in autoimmune islet infiltration, and besides promotion of IL-17 production, could directly regulate beta cell survival and function [11].

*Phylanthus emblica* L. (Figure 1A) is widely distributed in tropical and subtropical regions including Indian, China, Malay Peninsula, and Taiwan. *Phylanthus emblica* L. consists of numerous favorable active substances exhibiting health promotion functions including anti-tumor, anti-aging, anti-inflammatory, anti-bacterial, and lipid modulating activities [12].

The fruits of *P. emblica* have been shown to contain ascorbic acid, phenols (including ellagic acid and gallic acid), quercetin, kaempferol, corilagin, geraniin, and proanthocyanidins [13,14,15,16]. The roots of *P. emblica* have been shown to contain glycosides and tannins [6,11]. The major functional activities of *P. emblica* is due to its excellent antioxidant activity [14,16,17,18,19,20].

Previous studies have shown that the components of gallic and ellagic acid of *P. emblica* led to significant antioxidant activities and hepatoprotective effects [21,22]. *Phyllanthus emblica* extract has been shown to enhance these antioxidant activities [23]. Previously, *P. emblica* has been shown in animal studies to lower blood glucose levels in different models of diabetic rats [18,19,20].

The animal model of cyclophosphamide (Cyp)-accelerated auto-immune diabetes in non-obese diabetes (NOD) mice could raise autoimmunity by reducing regulatory T cell (Treg) lymphocyte numbers [24]. In the present study, we employed Cyp to rapidly induce diabetes with synchronization in NOD mice [25]. The preparation of ethyl acetate extract from *P. emblica* L. (EPE) was described as in Figure 1B. In the present study, we employed both models of spontaneous non-obese diabetes (S-NOD) and cyclophosphamide-accelerated NOD (Cyp-NOD) mice to clarify the antidiabetic and immunoregulatory activities of EPE and explore how EPE affects the β cells and modulates pro-inflammatory cytokines and Th1/Th2/Th17 cells. Furthermore, we investigated whether EPE has the protective effects of pancreas β cells by the up-regulation of the percentage (%) of Bcl-2/Bax and whether EPE has a therapeutic potential in the preventive effects of type 1 DM and immunoregulation. Finally, we attempted to find marker components of the extract, and then assess the improvement of activate insulin signaling using the individual seven fractions of the ethyl acetate from fruit extracts to assess its mechanism responsible for the functional activities in vitro.

## 2. Results

### 2.1. Animal Study

#### 2.1.1. Part I: Effects of Ethyl Acetate Extract from *Phyllanthus emblica* L. (EPE) in S-NOD T1D Mice

##### Body Weight, Relative Tissue Weight, Blood Glucose, HbA1_C_, Insulin, Adiponectin, and Leptin Levels

As shown in Figure 2A, for treatment with EPE for 15 weeks, the S-NOD+EPE mice showed a decrease in body weight compared with the S-NOD-Con group treated with EPE from week 4 to week 15 (Figure 2A). S-NOD+EPE mice showed a decrease in relative spleen weights compared with the S-NOD-Con group (*p* < 0.05, respectively) (Table 1). Due to the large variation between the two groups (in addition to the relative spleen weights), there was no change in the relative tissue weight compared with the S-NOD-Con group. Figure 2B shows the blood glucose levels of the initial time and the EPE-treated mice during the experimental 15 weeks. S-NOD+EPE mice had lowered blood glucose levels compared with the S-NOD-Con group at week 11 and week 15 (*p* < 0.05, *p* < 0.001, respectively) (Figure 2B). S-NOD+EPE mice had lowered blood HbA1_C_ concentrations compared with the S-NOD-Con group (*p* < 0.001) (Figure 2C). S-NOD+EPE mice showed a significant increase in blood insulin, adiponectin, and leptin levels compared with the S-NOD-Con group (*p* < 0.001, *p* < 0.001, *p* < 0.001, respectively) (Figure 2D–F).

##### Pro-Inflammatory Cytokines

Blood cytokine data show that EPE-treated S-NOD mice had decreased blood levels of IFN-γ, TNF-α, IL-1β, and IL-6 (*p* < 0.001, *p* < 0.001, *p* < 0.001, *p* < 0.001, respectively) (Figure 2G,H), but increased blood levels of IL-4, IL-10, and TGF-β1 (*p* < 0.001, *p* < 0.001, *p* < 0.001, respectively) by ELISA analysis compared with the S-NOD-Con group (Figure 2G,H).

##### Flow Cytometry

Flow cytometric data show that EPE-treated S-NOD mice had a decreased CD4^+^ subsets T cell distribution of CD4^+^IL-17 (Figure 3A), but an increased CD4^+^ subsets T cell distribution of CD4^+^IL-4 (Figure 3B) and CD4^+^ forkhead box P3 (Foxp3) (Figure 3D) compared with the S-NOD-Con group. No difference was observed on CD4^+^ interferon gamma (IFN-γ^+^) between the S-NOD+EPE group and the S-NOD-Con group (Figure 3C). EPE-treated S-NOD mice had a decreased percentage per 10,000 cells of CD4^+^IL-17, CD4^+^IL-4, and CD4^+^Foxp3 compared with the S-NOD-Con group (*p* < 0.001, *p* < 0.05, *p* < 0.05; respectively) (Figure 3E).

##### Target Gene Expression Levels in The Pancreas

The EPE-treated S-NOD group showed a marked decrease in the expression levels of interferon gamma (IFN-γ) and tumor necrosis α (TNF-α), but a significant increase in the expression levels of IL-4, IL-10, and transforming growth factor-β1 (TGF-β1) in the pancreas compared with the S-NOD-Con group (*p* < 0.001, *p* < 0.001, *p* < 0.001, *p* < 0.001, *p* < 0.001, respectively) (Figure 4A,B).

##### Insulin-Expressing β Cells, IRS Score on Glucagon and Insulin, and the Pancreatic Islet Numbers

Pathogenesis photographs revealed that administration of EPE to S-NOD mice caused an increase in pancreatic insulin-expressing β cells (brown) compared with S-NOD Con mice (*p* < 0.001) (Figure 5A). EPE-treated S-NOD mice showed a significant decrease in the average immunoreactive system (IRS) score of glucagon but an increase in the average IRS score of insulin within the pancreas, and an enhanced number of the pancreatic islets compared with the S-NOD Con group (*p* < 0.001, *p* < 0.001, *p* < 0.001, respectively) (Figure 5B).

##### The Percentage of Bcl-2/Bax

The green fluorescence/red fluorescence represents Bcl-2/Bax per cell under one vision. EPE-treated S-NOD mice showed a significant increase in the percentage of Bcl-2 (green)/Bax (red) compared with the S-NOD-Con mice by fluorescent staining analysis of islets counting by Image J software at 100× or 200× (*p* < 0.001, *p* < 0.001, respectively) (Figure 5C–E).

#### 2.1.2. Part II: Effects of EPE in Cyp-Induced T1D Mice

##### Body Weight, Relative Tissue Weight, Blood Glucose, HbA1_C_, Insulin, Adiponectin, and Leptin Levels

As shown in Figure 6, for the treatment with EPE for 4 weeks, EPE-treated mice showed a decrease in body weight at week 1 (*p* < 0.05), there was no difference in week 2, 3, or the final body weight (week 4) between the Cyp-NOD+EPE group and the Cyp-NOD-Con group (Figure 6A). Cyp-NOD+EPE mice showed both a decrease in relative tissue weights of spleen and skeletal muscle compared with the Cyp-NOD-Con group (*p* < 0.05, *p* < 0.05, respectively) (Table 2). As shown in Figure 6B, there was no difference in blood glucose levels between Cyp-NOD Con mice and Cyp-NOD+EPE mice at the initial time (week 0). At the final time (week 4), Cyp-NOD+EPE mice had significantly lower blood glucose levels compared with the Cyp-NOD Con group (*p* < 0.001) (Figure 6B). Cyp-NOD+EPE mice had significantly lower blood HbA1_C_ concentrations compared with the Cyp-NOD Con group (*p* < 0.001) (Figure 6C). Cyp-NOD+EPE mice showed a significant increase in blood insulin, adiponectin, and leptin levels compared with the Cyp-NOD-Con group (*p* < 0.001, *p* < 0.001, *p* < 0.001, respectively) (Figure 6D–F).

##### Pro-Inflammatory Cytokines

Blood cytokine data show that EPE-treated Cyp-NOD mice had decreased blood levels of IFN-γ, IL-1β, and IL-6 (*p* < 0.001, *p* < 0.01, *p* < 0.001, respectively), but increased blood levels of IL-4, IL-10, and TGFβ-1 (*p* < 0.001, *p* < 0.001, *p* < 0.001, respectively) by ELISA analysis compared with the Cyp-NOD Con group (Figure 6G,H).

##### Flow Cytometry

Flow cytometric data show that EPE-treated Cyp-NOD mice had a decreased CD4^+^ subsets T cell distribution of CD4^+^IL-17 and CD4^+^IFN-γ (Figure 7A,B), but an increased CD4^+^ subsets T cell distribution of CD4^+^IL-4 and CD4^+^Foxp3 compared with the Cyp-NOD Con group (Figure 7C,D). EPE-treated Cyp-NOD mice had a decreased percentage per 10,000 cells of CD4^+^IL-17 and CD4^+^IFNγ, and increased CD4^+^IL-4 and CD4^+^Foxp3 compared with the Cyp-NOD Con group (*p* < 0.001, *p* < 0.05, *p* < 0.05, *p* < 0.05, respectively) (Figure 7E).

##### Target Gene Expression Levels in The Pancreas

The EPE-treated Cyp-NOD group showed a marked decrease in the expression levels of IFN-γ and TNF-α, but a significant increase in the expression levels of IL-4, IL-10, and TGF-β1 in the pancreas compared with the Cyp-NOD-Con group (*p* < 0.001, *p* < 0.001, *p* < 0.001, *p* < 0.001, *p* < 0.001, respectively) (Figure 8A,B).

##### Insulin-Expressing β Cells, IRS Score on Insulin, and the Pancreatic Islet Numbers

Pathogenesis photographs revealed that administration of EPE to Cyp-NOD mice caused an increase in pancreatic insulin-expressing β cells (brown) compared with Cyp-NOD Con mice (*p* < 0.001) (Figure 9A). EPE-treated Cyp-NOD mice had a significantly increased average IRS score of insulin within the pancreas, and an enhanced number of the pancreatic islet numbers compared with the Cyp-NOD Con group (*p* < 0.001, *p* < 0.01, respectively) (Figure 9B).

##### The Percentage of Bcl-2/Bax

EPE-treated Cyp-NOD mice showed a significant increase in the percentage of Bcl-2/Bax compared with the Cyp-NOD Con group by fluorescent staining analysis of islets at 100× or 200× (*p* < 0.001) (Figure 9C–E).

### 2.2. Preparation of Seven Fractions of EPE

#### 2.2.1. Seven Fractions of EPE on Targeted Gene Expressions In Vitro

The preparation procedure of seven fractions of EPE was described in Figure 10A. Seven fractions of EPE (EA) are described as EA. EA included 2~10% EA fraction 1 (EA-1), 10~20% EA fraction 2 (EA-2), 20~30% EA fraction 3 (EA-3), 30~50% EA fraction 4 (EA-4), 50~70% EA fraction 5 (EA-5), 70~100% EA fraction 6 (EA-6), and 100% EA fraction 7 (EA-7). Glucose transporter 4 (GLUT4) is known to play a central role in blood glucose homeostasis [26]. Either the stimulation of insulin or contraction facilitates glucose uptake by translocate GLUT4 to the cell membrane [27,28]. The mechanisms of promoting glucose uptake into skeletal muscle include the insulin-dependent mechanisms leading to activation of Akt/PKB and contraction-regulated stimulation [29,30]. Therefore, we chose membrane GLUT4 and activation of Akt/PKB as anti-diabetic target genes. Figure 10B,C show that the insulin-, EA-3-, EA-4-, EA-5-, EA-6-, and EA-7-treated groups had increased expression levels of membrane GLUT4 or raised Akt activation (phospho-Akt/total-Akt; p-Akt/t-Akt) in comparison to the control group. Our findings show that EA-6 displays the best activity of phospho-Akt/total-Akt and expression levels of GLUT4 in vitro. This study was then designed to explore the marker ingredient of EPE responsible for the anti-diabetes.

#### 2.2.2. Analysis of EA-6 Fraction

Polyphenolic compounds are natural antioxidants that play a major functional role in plants by scavenging free radicals [31]. Polyphenolic compounds can be divided into two large categories: flavonoids and phenolic acids.

This study was designed to examine whether polyphenolic compounds were found in the EA-6 fraction or not. The retention time of gallic acid was 4.3 min as a reference compound to analyze the EA-6 fraction qualitatively and quantitatively. As shown in Figure 10E, the major absorption peaks of phenolic acids were consistent with the Photo Diode Array (PDA) spectra of the reference materials.

As shown in Figure 10E, the main ingredient in the phenolic compound is gallic acid found in EA-6 (33.4%).

## 3. Discussion

The present study showed that administration of EPE to both S-NOD mice and Cyp-NOD mice had significantly lower blood glucose HbA1_C_ levels and decreased insulitis scores by HE staining compared with the S-NOD-Con and Cyp-NOD-Con mice, respectively. Since the primary etiology of type 1 diabetes mellitus (T1D) is the pancreas β cell suffering from autoimmune aggressivity and selective destroy [32], the key point in preventing and ameliorating T1D is to protect the pancreas β cells from being destroyed. Therefore, EPE had ameliorated T1D and decreased insulitis effects, which is associated with a decrease in the specific destruction of pancreas β cells.

Apoptosis is the major form of pancreas β apoptosis in NOD mice. The pancreas β cells of 15-week-old NOD mice have shown to be the peak of abnormal apoptosis [33]. Apoptosis plays a key role in the regulation of the Bcl-2 gene family in the occurrence and development of pancreas β cell apoptosis. The Bcl-2 gene family could protect cells from the occurrence of apoptosis, whereas the Bax gene could promote apoptosis. Bcl-2/Bax ratios are demonstrated in the basis of regulation of cell apoptosis [34]. The present results showed that administration of EPE to both S-NOD and Cyp-NOD mice increased the expression levels of Bcl-2 but decreased the expression levels of Bax, and thus contributed to the increased percentage (%) of Bcl-2/Bax compared with the S-NOD Con mice and Cyp-NOD Con mice by immunofluorescence staining assays. Our findings imply that EPE displayed the protective effects of pancreas β cells through the regulation of the percentage (%) of Bcl-2/Bax.

CD4^+^ T lymphocytes play the core role in the disease process of type 1 diabetes mellitus. CD4^+^ T lymphocytes include at least Th1 lymphocytes and Th2 subsets. Previous studies have shown that cell function disequilibrium of Th1 cells and Th2 cells exists [35]. Th1 cells principally secrete numerous cytokines including IFN-γ, IL-1, IL-2, and TNF-α, and whereas Th2 cells principally secrete IL-4, IL-10, and TGF-β1 cytokines. During the disease process of type 1 diabetes mellitus, the function of the destructive Th1 cells and cytokines within the pancreas β cells had enhanced, whereas the function of the protective Th2 cells and cytokines had decreased [36]. Since different cytokines in the pancreas β cells exhibited different destructive effects, we could bypass stimulating the production of Th2 cell cytokines, and inhibiting production of Th1 cell cytokines to lead to the regulation of the equilibrium of Th1/Th2 cytokines, and thus contribute to the goal of the treatment of type 1 diabetes mellitus. Our results for the S-NOD mice (in Part I) showed that EPE-treated S-NOD mice showed a decrease in the pancreatic expression levels of IFN-γ and TNF-α which led to reduced blood levels of IFN-γ and TNF-α cytokines, which are associated with Th1 cell cytokines, but showed an increase in the expression levels of IL-4, IL-10, and TGF-β1 (with relation to Th2 cell cytokines) which led to enhanced blood levels of IL-4, IL-10, and TGF-β1 cytokines compared with S-NOD Con mice. Our results for the Cyp-NOD mice (in Part II) showed that EPE- treated Cyp-NOD Con mice showed a decrease in the pancreatic expression levels of IFN-γ which led to lower blood IFN-γ cytokine levels, but an increase in the pancreatic expression levels of IL-4, IL-10, and TGF-β1 which led to enhanced blood levels of IL-4, IL-10, and TGF-β1 cytokines compared with Cyp-NOD Con mice. These results imply that EPE could prevent or ameliorate the occurrence of type 1 diabetes mellitus through the regulation of the Th1/Th2 ratio equilibrium, possibly due to one of the immunoregulatory mechanisms.

Our findings demonstrated the preventive effects that EPE had on the antidiabetic and immunoregulatory activities both in S-NOD and Cyp-NOD mice. The present study showed that administration of EPE to both S-NOD and Cyp-NOD mice decreased insulitis and lowered the incidence of T1D and its underling mechanism of actions due to the transformation from Th1 and its cytokines to Th2 and its cytokines, and thus contributed to an increase in ratios (%) Bcl-2/Bax to avoid the destruction of pancreas β cells.

Our findings demonstrated that administration of EPE to both S-NOD and Cyp-NOD mice decreased the incidence of T1D and reduced insulitis scores, with an increase in (green) immunofluorescence staining of Bcl-2 but a decrease in (red) immunofluorescence staining of Bax compared with both S-NOD and Cyp-NOD mice. Moreover, administration of EPE to both S-NOD and Cyp-NOD mice decreased the expression levels of TNF-α and IFN-γ, and markedly decreased the severity degrees of insulitis and immunoreactive (IRS) scores to lower the incidence of diabetes. In the present study, EPE treatment not only prevented T1D but also decreased the inflammatory response within the pancreas with relation to a decrease in specific destruction within pancreas β cells, with its underling mechanisms possibly owing to the regulation of Th1/Th2 cells cytokine immune disequilibrium.

Recent studies have shown that IL-17 plays a core role in the process of various types of chronic inflammation including rheumatoid arthritis, multiple sclerosis, and systemic lupus erythematosus with an increase in expression levels of IL-17 [37]. Evidence has shown that blood glucose levels increase as the incidence of diabetes increases, and the numbers of Th17 cell subsets increase, implying that Th17 plays a determinant role in disease incidence and formation of diabetes mellitus.

Th17 cells have been shown to play a key role in the incidence of autoimmune response [38]. Previous studies have shown that the number of Th17 cells increases as the glucose levels in NOD mice increase, implying that the Th17 cell plays a role in the incidence of type 1 diabetes mellitus [39]. The forkhead box protein 3 (Foxp3) is primarily expressed in a subset of CD4^+^-T cells that play an inhibitory part in the immune system [40]. Numerous factors including cytokines modulate the production of Foxp3 T cells [40]. Cytokines work together with TGF-β1 and promote the production of Foxp3 T cells [40]. Foxp3+ T cells are reported to have therapeutic potential in the treatment of inflammatory diseases [40]. Our findings showed that EPE treatment in both S-NOD and Cyp-NOD mice lowered blood glucose levels and increased the number of Foxp3 subsets of CD4^+^ T cells and reduced blood cytokine levels of TGF-β1, suggesting that EPE has potential protective effects in T1D.

Evidence has demonstrated that T1D in humans displays an increase in the production of T cell IL-17 [41]. One study showed that T1D children displayed an increase in IL-17 secretion from both CD4 and CD8 T cells [42]. Both IL-6 and IL-1β were demonstrated to provide Th17 development [43,44], and thus the researchers concluded that the monocytes of T1D individuals displayed an increase in the mRNA levels of both IL-6 and IL-1β, which implied an explanation for the increase in IL-17 production [45]. T helper type 1 (Th1) lymphocytes and their hallmark cytokine IFN-γ are central in T1D pathogenesis [2]. Our findings showed that EPE treatment in Cyp-NOD mice lowered blood glucose levels and decreased both the numbers of IL-17 CD4^+^ and reduced the pancreatic expression levels of IFN-γ, and reduced blood IL-6, IL-1β, and IFN-γ cytokine levels, implying that EPE regulates the interaction of Th1 and Th17 cells which leads to decreased proinflammatory cytokine IL-6 and IL-1β levels, thus contributing to a steady state of immunoregulation.

In line with previous studies [7,41,42,43,44,45,46], the present study showed that with the administration of EPE to both S-NOD and Cyp-NOD mice, blood glucose levels were lowered, and the number of CD4^+^ IL-17 cell distribution per 10,000 cells (%)decreased, but the number of CD4^+^ IL-4 and CD4^+^Foxp3^+^ increased, and blood levels of IL-1β and IL-6 were reduced, but TGF-β1 levels increased. Th17 cells were demonstrated to secrete IL-6 and IL-17, which play a part in inflammatory and autoimmune diseases. Therefore, the present study indicated that EPE could ameliorate the incidence of type 1 diabetes mellitus and immunoregulation accompanied by a decrease in the number of CD4^+^ IL-17 cells and blood cytokine levels of IL-6 through the regulation of Th17 cells.

Th1 cells could produce cytokine IFN-γ to inhibit the differentiation of the Th17 cell, whereas the Th2 cell produces cytokines to inhibit the differentiation of the Th17 cell. Taking the administration of EPE to both S-NOD and Cyp-NOD mice together, blood glucose levels lowered, possibly due to the mechanisms of regulation of Th1 cell differentiation and the indirect modulation of numerous cytokines which impactthe mutual restraint and interaction of Th1, Th2, Treg cell, and Th17 cell. The convergent effect of EPE is to regulate and mutually restrain T cell subsets, and thus contribute to immunoregulation equilibrium.

Our findings show that the administration of EPE to S-NOD and Cyp-NOD mice increased blood levels of adiponectin and leptin. Wu et al. demonstrated that treatment with the globular domain of adiponectin increased glucose uptake [47], suggesting that EPE could regulate the secretion of adiponectin and lead to an increase in glucose uptake, and in turn control glucose homeostasis. Evidence has shown that leptin could substitute for insulin to control blood sugar fluctuations in patients with type 1 diabetes [48]. Nevertheless, the mechanism of action still remains unknown. This study demonstrated that EPE had favorable effects on leptin levels, implying that EPE plays a critical role in glucose metabolism partly by enhancing blood leptin levels.

Skeletal muscle is the major tissue responsible for insulin-mediated glucose utilization [49]. Due to its expression of GLUT 4 representative to human skeletal muscle cells, the C2C12 myoblast cell line is employed in vitro. Since membrane GLUT4 and p-Akt/t-Akt play critical roles in type 1 diabetes, the cell culture analyses of expressions of GLUT4 and p-Akt/t-Akt were determined to clarify the anti-diabetic activity using Western blotting. Our findings showed that EA-6 displayed the best activity out of the seven fractions for increased expression levels of the membrane GLUT4 and phospho-Akt/total-Akt. We then attempted to find the marker ingredient using HPLC analysis.

Our HPLC analytic data showed that EPE extraction yield was verified in Figure 10E, showing that the extraction percentage of total polyphenol gallic acid (>33.4%) of the EA-6 fraction is the major component. Gallic acid, ellagic acid, and chebulagic acid have been found to be antioxidant [50]. Gallic acid has been found to display anti-inflammatory activity by regulating inflammatory cytokines such as TNF-α, TGF-β1, IL-1β, and IL-6 [51]. Ellagic and gallic acid were demonstrated to exert anti-inflammatory potential with relation to IL-6 cytokine [52]. Gallic acid and other acids have been found to have an immunosuppressive effect on cytotoxic T lymphocyte-mediated cytotoxicity [53]. One of our significant findings was that the administration of EPE to both S-NOD and Cyp-NOD mice decreased the number of CD4^+^ IL-17 cell distribution per 10,000 cells (%) but increased the number of CD4^+^ IL-4 and CD4^+^Foxp3^+^ by inhibiting Th1 cells and stimulating Th2 cells, and regulating Th17 cells convergently to secrete pro-inflammatory cytokines (including a decrease in blood IL-1β and IL-6 and an increase in TGF-β1 levels) and had an impact on the mutual restraint and interaction of Th1, Th2, and Th17 cells. Another significant finding was that EPE lowered blood glucose and HbA1_C_ due to EPE’s antioxidant activity in the blood and within the pancreas, and moreover, EPE increased the expression levels of Bcl-2 but decreased Bax, and increased Bcl-2/Bax (%) by IF stain, thus EPE could protect the pancreas β from apoptosis and damage in two NOD mice models. An explanation for EPE’s antidiabetic and immunoregulatory activity is that the major constituents of fruit extract contain total polyphenol component gallic acid and display the majority of antioxidant, anti-diabetic, anti-inflammatory, and immunosuppressive activity, and the indirect modulation of pro-inflammatory cytokines and mutual restraint and interaction of Th1, Th2, and Th17 cells, thus contributing to the preventive effects of EPE on type 1 DM and immunoregulation.

## 4. Materials and Methods

### 4.1. Chemicals

Anti-mouse CD4 FITC (Clone CT-CD4) (rat IgG2a) (no. MBS520110) was obtained from MyBioSource.com (San Diego, CA, USA) accessed on 7 April 2022. Anti-mouse IL-17 PE L-25R (IL-17RB) monoclonal antibody (MUNC33) (no. 12-7361-82), IL-4 PE (IL-4 monoclonal antibody) (11B11) (no. 12-7041-41), IFN-γ PE (IFN gamma monoclonal antibody) (XMG1.2) (no. 12-7311-82), and Foxp3 PE (FOXP3 monoclonal antibody) (NRRF-30) (no. 12-4771-82) were obtained from eBiosciences (San Jose, CA, USA). IL-17 (TC11-18H10) rabbit anti-rat monoclonal antibody (no. sc-52567) was obtained from Santa Cruz Biotechnology (Santa Cruz, CA, USA). Western blotting antibodies to TGF β-1 (no. MA5-15065), TNF-α (no. 710162), and IFN γ (no. 701121) were obtained from Invitrogen Inc. (Carlsbad, CA, USA), and IL-10 (no. MBS8238272) and IL-4 (no. MBS822213) were obtained from MyBioSource, Inc. (San Diego, CA, USA).

### 4.2. Fruit Materials

Fruits of *Phyllanthus emblica* L. (Figure 1A) were purchased from Taiwan Miaoli County, Ogan marketing Cooperation. These fruits were identified by China Medical University, Taiwan, and the voucher specimen (CMPF385) is replaced.

### 4.3. Preparation of Ethyl Acetate Extract from Phyllanthus emblica L. (EPE)

The 7.3 kg of *Phyllanthus emblica* L. fruit (Figure 1A) was dried and pulverized into 1.34 kg fine powder. The fine powder was extracted with 6.3 L methanol three times at room temperature to concentration under vacuum, and then the crude methanolic extract (331.86 g) was obtained. The crude methanolic extract was subjected to suspension in H_2_O and partition with EtOAc three times, respectively, followed by concentration under reduced pressure, and then the H_2_O fraction (323.71 g) and the EtOAc fraction (47.28 g) were obtained (Figure 1B). The EtOAc fraction was employed for the animal study.

### 4.4. Animal Treatments

#### 4.4.1. Part I: EPE or Vehicle Administered to S-NOD Mice

Fourteen female NOD/ShiLtJNarl mice (3 weeks old) were purchase from National Laboratory Animal Breeding Center and then housed in a pathogen-free environment. The animal study was approved according to the guidelines of Central Taiwan University of Science and Technology, Taiwan, in accordance with the National Institutional Animal Care and Use Committee and approved by local animal ethics committee (Animal Ethics Committee, permit no. 109-CTUST-007). As our previous report described [54], this regulation described the early euthanasia/humane endpoints for animals who became severely ill to reduce pain to the minimum degree. The clinical signs employed to decide when to euthanize the animals were the following: 1. reduced body weight by loss of 25% of original body weight; 2. for rodents: reduced food intake within 3 days below 50% of normal ingestion; 3. weak or dying status; 4. organs of animals with severe loss of functions and with clinical symptoms; 5. tumor; 6. pain that could not be controlled following treatment with analgesics [54]. After a one-week waiting period, the mice were randomly divided into two groups. One group (*n* = 7) of mice was the control group (spontaneous NOD-Con (S-NOD-Con); Group I: normal mice) and were orally given identical volumes of vehicle. The other group of mice (*n* = 7) (S-NOD+EPE) (Group II: EPE-treated S-NOD mice) were orally given EPE at a dosage of 400 mg/kg body weight once daily for 15 weeks.

#### 4.4.2. Part II: EPE or Vehicle Administered to Cyp-NOD Mice

Twenty-four female NOD/ShiLtJNarl mice (3 weeks old) were purchase from National Laboratory Animal Breeding Center and then housed in a pathogen-free environment. The animal study was approved by the school Animal Ethics Committee (no. 109-CTUST-007). After a one-week adaptation period, all mice were randomly divided into two groups including the (Group I) Cyp-NOD-Con and the (Group II) Cyp-NOD+EPE group. To accelerate diabetes, we used cyclophosphamide (Cyp, Pharmacia, North Ryde, Australia). Cyp-induced diabetes was performed by injecting 200 mg/kg body weight intraperitoneally (i.p.) twice 14 days apart in female NOD mice. Mice were orally given EPE from the first Cyp injection (at day 0) once daily for 28 days to the end of the experiment. EPE or vehicle administered to Cyp-NOD 4-week-old female mice at a dose of 400 mg/kg body weight once daily for 4 weeks. After treatment with EPE for 15 weeks or 28 days, the mice (after 10 h of fasting) were sacrificed, and peripheral tissues were weighed. Parts of tissues were immediately stored at −80 °C for targeted gene analysis. Blood glucose analysis for cytokine (including insulin, adiponectin, and leptin) levels and body weights were performed as per previous reports [54,55,56]. At the end, blood samples were collected for the analyses of levels of blood glucose, insulin, and HbA1c, and the pancreas and spleen tissues were dissected for the analyses of histology and immunofluorescence staining (including expression levels of Bcl and Bax), the expression levels of targeted genes by Western blotting (including of the expression levels of TNF-α, IFN-γ, and IL-10), and Th1/Th2/Th17/Treg cell distribution by flow cytometry.

#### 4.4.3. Measurements of Blood Glucose, HbA1_C_, Insulin, Adiponectin, and Leptin Levels

Blood samples were obtained from the retro-orbital sinus of 10 h fasted mice. Blood glucose levels were analyzed using the glucose oxidase method. Insulin, leptin, and adiponectin levels were measured using enzyme-linked immunosorbent assay (ELISA) kits as per previous procedures [54,55,56].

#### 4.4.4. Assessment of Pro-Inflammatory Cytokines

Blood pro-inflammatory cytokine levels (including IL-10, IL-4, IL-6, IL-1 β/IL-1F2, IFN-γ, TNF-α, and TGF-β1) were measured using enzyme-linked immunosorbent assay (ELISA) kits from R&D Systems, Inc. (Minneapolis, MN, USA). The detailed ELISA kits’ brane numbers were mouse IL-10 Quantikine ELISA kit (no. M1000B-1), mouse IL-4 Quantikine ELISA kit (no. M4000B), mouse IL-6 Quantikine ELISA kit (no. M6000B), mouse IL-1 β/IL-1F2 Quantikine ELISA kit (no. MLB00C), mouse IFN-γ Quantikine ELISA kit (MIF00), mouse TNF-α Quantikine ELISA kit (no. MTA00B), and mouse TGF-β 1 DuoSet ELISA (no. DY1679-05), respectively.

#### 4.4.5. Flow Cytometry (Fluorescence Activated Cell Sorting; FACS)

The harvested spleens of mice were firstly teased with forceps and put into 6-well plates. The excised spleen pieces were incubated for 30 min at 37 °C with 1 mL digestion buffer containing Collagenase IV (100 U/mL) (Roche, Basel, Switzerland) and Dnase I (20 µg/mL) (Roche) solution with 1% FBS (Gibco,New York, NY, USA). Then, the cell suspension was pressed through the 70 μm strainer and then washed with cell staining buffer. To remove the RBC in pellets, 5 mL RBC lysis buffer was added and incubated for 5 min at 37 °C. Samples were centrifuged for 5 min at 350× *g* at 4 °C, then the supernatants were removed. Pellets were fixed by 4% paraformaldehyde (PFA) (Biolegend, San Diego, CA, USA) and permeabilized by permeabilization buffer (Biolegend) for intracellular targeted proteins staining. Samples were subdivided into 4 groups and stained with CD4-APC, IL4-PE; CD4-APC, IL17-PE; CD4-APC, IFNγ-PE; and CD4-APC, Foxp3-PE (Elabscience); respectively. Samples were processed and analyzed using a Novocyte 3000 instrument (Agilent Technologies, Richmond, BC, Canada).

#### 4.4.6. Histological Examination and Immunohistochemical (IHC)

The pancreata were fixed in 10% formaldehyde and embedded in paraffin. At least 10 islets from each mouse were scored for immunoreactive system (IRS) according to both the intensity and quantity of the dyeing signal for makers (including glucagon and insulin) within the pancreas using the following criteria of IRS score: 0–1: negative; 2–3: mild positive; 4–8: moderate positive; 9–12: strongly positive [57,58]. A (quantity) included the following issues: 0 = no positive cells, 1 ≤ 10% of positive cells, 2 = 10–50% positive cells, 3 = 51–80% positive cells, and 4 ≥ 80% positive cells. B (intensity) included the following conditions: 0 = no color, 1 = mild positive, 2 = moderate positive, and 3 = strong positive. The IRS score was a multiplication of A and B. Final IRS score = A (quantity) × B (intensity) = 0–12 [58]. Furthermore, the number of pancreatic islets was also calculated within the pancreas by a pathology veterinarian. The morphological procedure was performed as per previous studies [54,55,56]. Insulin-expressing β cells were stained for brown and glucagon-expressing α-cells for green by IHC in the pancreatic islets as per previous studies [54,55,56].

#### 4.4.7. Immunofluorescence Staining (IF)

The tissue slide was dewaxed and dewatered, soaked in 3% H_2_O_2_ in methanol for 10 min, soaked in citrate buffer 95 °C for 10 min, and then cooled down to 50 °C. The slide was washed twice with 1X PBS (phosphate-buffered saline, pH:7.4) for 5 min each time, and the surrounding redundant water was wiped clean (tissues were still kept wet). Then, it underwent the reaction according to the immunofluorescence staining kit manual (TATS02F; BIOTnA; Taiwan). Briefly, the tissue was flooded with 200 μL fluorescence blocking reagent and reacted at room temperature for 1 h, and then washed with1X PBS twice, and then diluted with anti-Bcl-2 antibody (C-2) (sc-7382; Santa Cruz Biotechnology, Inc.) and anti-Bax Antibody (B-9) (sc-7480; Santa Cruz Biotechnology, Inc.), and then 200 μL of the diluted antibody was added and flooded the tissue at 4 °C overnight. Then, the slide was washed with 1X PBS twice for 5 min each time and the surrounding redundant water was wiped clean (tissues were still kept wet). Then, the diluted secondary antibody anti-rabbit IgG (H+L)-488 and goat anti-mouse IgG (H+L)-594 (TATS02F; BIOTnA; Taiwan) were added, and 200 μL of the diluted antibody was added and flooded the tissue at room temperature avoid light for 1 h. Then, it was washed with 1X PBS twice for 5 min each time and the surrounding redundant water was wiped clean (tissues were still kept wet), and the DAPI (Applied Biological Materials Inc. (abm); Richmond, BC, Canada) was added to perform cellular nuclear staining and mounting, and the image was taken with the fluorescence microscopy (OLYMPUS model: BX53) and the fluorescence distribution within the cells was observed. The percentage (%) of Bcl-2/Bax was calculated using the method of each group. Three slides under nine visions were observed and the cell numbers were counted by Image J software. The green fluorescence/red fluorescence under one vision was presented as Bcl-2/Bax per cell, and the number of Bcl-2/Bax per 100 cells was calculated as % of Bcl-2/Bax.

#### 4.4.8. Western Blotting Assay

The pancreas was immediately removed and quickly homogenized with RIPA buffer (SIGMA, Louis, MO, USA) prior to the Western blot [54,55,56]. The expression levels of the target genes were assessed by Western blot as depicted in previous reports [46,47,48] using the following antibodies. These were antibodies TGF β-1 (no. MA5-15065), TNF-α (no. 710162), and IFN-γ (no. 701121) from Invitrogen Inc. (Carlsbad, CA, USA), and IL-10 (no. MBS8238272) and IL-4 (no. MBS822213) from MyBioSource, Inc. (San Diego, CA, USA). Goat anti-mouse IgG was coupled to HRP secondary antibody (Jackson Lab., Inc., West Grove, USA) [54,55,56]. Finally, these results were detected by chemiluminescence kits (Amersham Biosciences ECL^TM^, Buckinghamshire, UK).

#### 4.4.9. Preparation of Seven Fractions of EPE

The preparation of seven fractions of EPE is depicted in Figure 10A. The fruits of *Phyllanthus emblica* (126.3 g) were extracted with methanol at 25 °C (3 × 7 d). The methanol extract was dried under vacuum and the remaining (31.4 g) was obtained. This was suspended in H_2_O and partitioned with EtOAc. The EtOAc-soluble fraction (5.6 g) was performed using column chromatography on silica gel with the gradient solvent systems of *n*-hexane and EtOAc as a mobile phase and seven fractions were obtained.

#### 4.4.10. Analysis of Seven Fractions of EPE on Targeted Gene Expressions In Vitro

Since membrane GLUT4 and p-Akt/t-Akt play critical roles in type 1 diabetes, the cell culture analyses of expressions of GLUT4 and p-Akt/t-Akt were determined to clarify the anti-diabetic activity using Western blot with antibodies specifically as those above. Briefly, the treatments of cells with seven fractions of ethyl acetate from *P. emblica* (including EA-1~EA-7) or insulin were initiated on day 6 as per our previous publication [59].

#### 4.4.11. HPLC Analysis

##### Fingerprint Analysis by HPLC

The analysis was performed on a HITACHI high-performance liquid chromatographic (HPLC) L-5000 system equipped with a degasser, pumps, and a photodiode array detector linked to a PC computer running the software program HPLC LACHROM version 1.0. For HPLC analysis, an aliquot (10 μL) was injected into the columns and eluted at 40 °C. The analytical column (250 × 4.6 mm i.d., 5 μm) used Thermo Hypersil GOLD C_18_ (Waltham, MA, USA) and the detection wavelength. For photodiode array detection, the wavelengths of standard compounds at their respective maximum absorbance wavelengths can be monitored at the same time. Identification was based on retention times and online spectral data in comparison with authentic standards.

##### Determination of Phenolic Compounds

The method used was in accordance with a previous report [60]. The mobile phase contained acetonitrile (solvent A) and acidified water with trifluoroacetic acid (0.05%, solvent B). The gradient program was as follows: 10–20% A (10 min), 20–28% A (5 min), 28–50% A (5 min), and 50–50% A (4 min). The flow rate was 1.0 mL/min, and the injection volumes of standards and samples were 10 μL. Identification was based on retention times and photo diode array (PDA) spectra as compared with commercial standards.

### 4.5. Statistical Analysis

These results were calculated by SPSS software (25.0.0.0, SPSS Inc., Chicago, IL, USA) with the analysis of non-parametric with the Kruskal–Wallis H test, followed by the Mann–Whitney *U* test. These results are presented as the mean and standard error. Whenever the *p* value was less than 0.05, this was taken as significant.

## 5. Conclusions

Our results showed that EPE-treated NOD mice or Cyp-accelerated NOD mice displayed a decrease in the levels of blood glucose and HbA1c, but an increase in blood levels. ELISA analysis showed that EPE decreased the blood levels of IFN-γ and TNF-α via Th1 cells and reduced IL-1β and IL-6 by Th17 cells but increased IL-4, IL-10, and TGF-1β via Th2 cells in both mice models. Western blotting analyses of the pancreatic target gene expressions showed that EPE inhibited the expression levels of inflammatory cytokines, including IFN-γ and TNF-α via Th1 cells but up-regulated the expression levels of IL-4, IL-10, and TGF-1β via Th2 cells in both mice models. In both mice models, histological examination of the pancreas revealed that EPE-treated mice had increased pancreatic insulin-expressing β cells (brown), the average IRS score of insulin and the numbers of the pancreatic islets, and an increased percentage (%) of Bcl-2/Bax to avoid the destruction of pancreas β cells. Flow cytometric data showed that EPE-treated mice in both mice types showed a decrease in the number of CD4^+^ IL-17 cell distribution per 10,000 cells (%), but an increase in the numbers of CD4^+^ IL-4 and CD4^+^Foxp3^+^ via the inhibition of Th1 cells and stimulation of Th2 cells. These data also showed the regulation of Th17 cells converged to secrete pro-inflammatory cytokines (such as lowering blood levels of IL-1β and IL-6, but increasing TGF-β1) and an impact on the mutual restraint and interaction of Th1, Th2, and Th17 cells, thus contributing to the preventive effects of EPE on TID and immunoregulation.

In conclusion (Figure 11), these results imply that EPE inhibits the development of autoimmune diabetes by regulating cytokine expression. EPE displays an improvement in pancreas immunoreactive scores and a decrease in pro-inflammatory cytokines. EPE exerts blood-glucose-lowering effects by regulating IL-17 expression. Our results demonstrated that EPE has therapeutic potential in immunoregulation and the prevention of T1D.

## Figures and Tables

**Figure 1 ijms-24-09922-f001:**
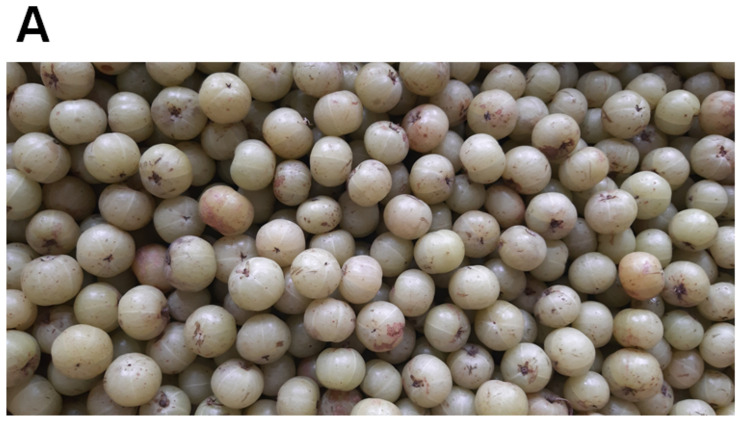
(**A**) The fruits of *Phyllanthus emblica* L. (**B**) Preparation of ethyl acetate extract from *Phyllanthus emblica* L. (EPE). The fruits of *Phyllanthus emblica* were extracted with methanol three times at room temperature to concentration under vacuum, and then the crude methanolic extract was obtained. The crude methanolic extract was subjected three times to the of suspension in H_2_O and partition with EtOAc, respectively, and followed by concentration under reduced pressure, and then the H_2_O fraction and the EtOAc fraction were obtained (**B**).

**Figure 2 ijms-24-09922-f002:**
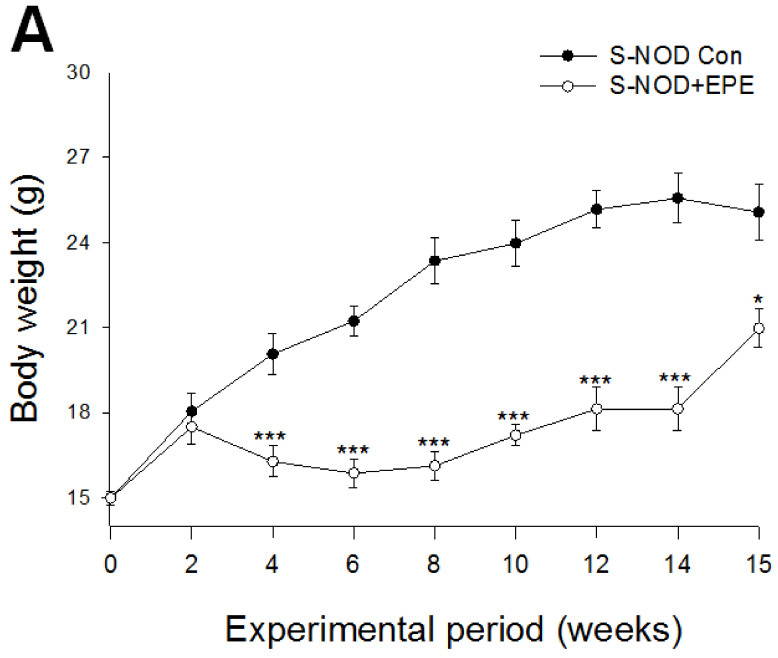
Effects of ethyl acetate from *Phyllanthus emblica* L. (EPE) on (**A**) body weight, (**B**) blood glucose, (**C**) HbA1_C_, (**D**) insulin, (**E**) adiponectin, (**F**) leptin levels, (**G**) blood cytokine levels-1: IL-4, IL-10, and IFN-γ, and (**H**) blood cytokine levels-2: IL-1β, IL-6, TGF-1β, and TNF-α levels in spontaneous non-obese diabetes (S-NOD) mice. * *p* < 0.05, or *** *p* < 0.001 compared to the S-NOD plus vehicle (S-NOD Con) group. All values are means ± SE (*n* = 7 per group). EPE: 400 mg/kg body weight.

**Figure 3 ijms-24-09922-f003:**
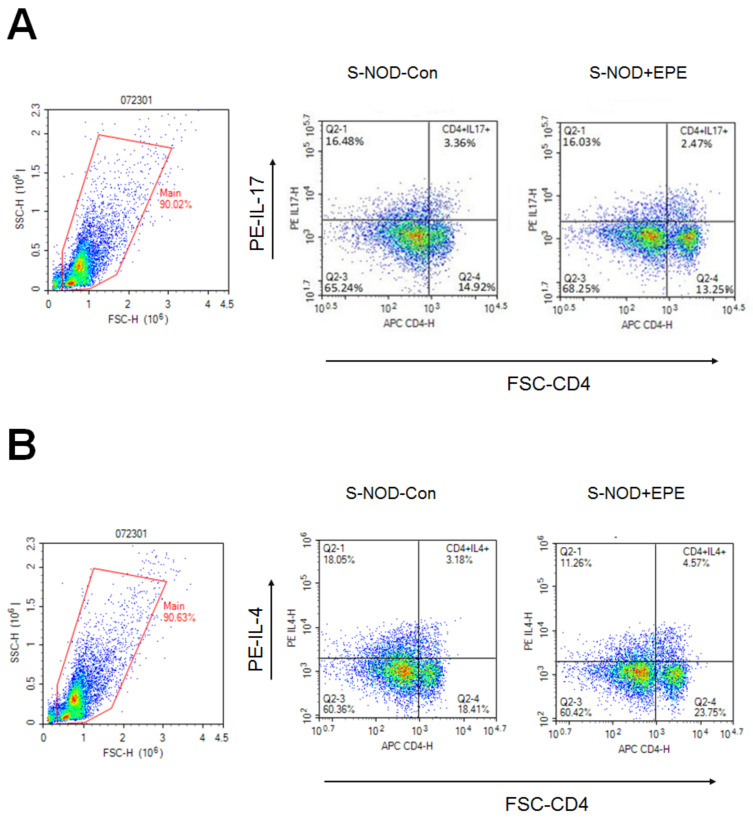
Flow cytometric data of ethyl acetate from *Phyllanthus emblica* L. (EPE) on numbers of (**A**) IL-17, (**B**) IL-4, (**C**) IFNγ, (**D**) Foxp3 subsets of CD4^+^ T cell distribution, and (**E**) four subset CD4^+^ T cell distribution per 10,000 cells (%) in spontaneous non-obese diabetes (S-NOD) mice. * *p* < 0.05 or ** *p* < 0.01 compared to the S-NOD plus vehicle (S-NOD Con) group. All values are means ± SE (*n* = 7 per group). EPE: 400 mg/kg body weight.

**Figure 4 ijms-24-09922-f004:**
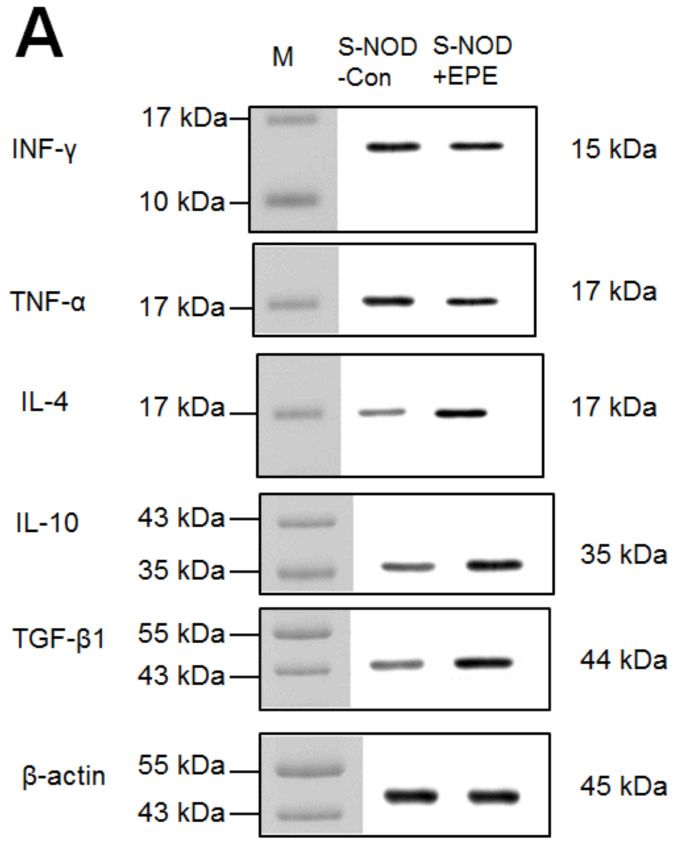
Target gene expression levels (including IFN-γ, TNF-α, IL-4, IL-10, or TGF-β1) in the pancreas following treatment of ethyl acetate extract from *Phyllanthus emblica* L. (EPE) in spontaneous non-obese diabetes (S-NOD) mice by Western blotting. (**A**) Representative image; (**B**) target gene expression quantification to β-actin. Protein was separated by 12% sodium dodecyl sulfate polyacrylamide gel electrophoresis (SDS-PAGE) detected by Western blotting. *** *p* < 0.001 compared to the S-NOD plus vehicle (S-NOD Con) group. All values are means ± SE (*n* = 7 per group). EPE: 400 mg/kg body weight.

**Figure 5 ijms-24-09922-f005:**
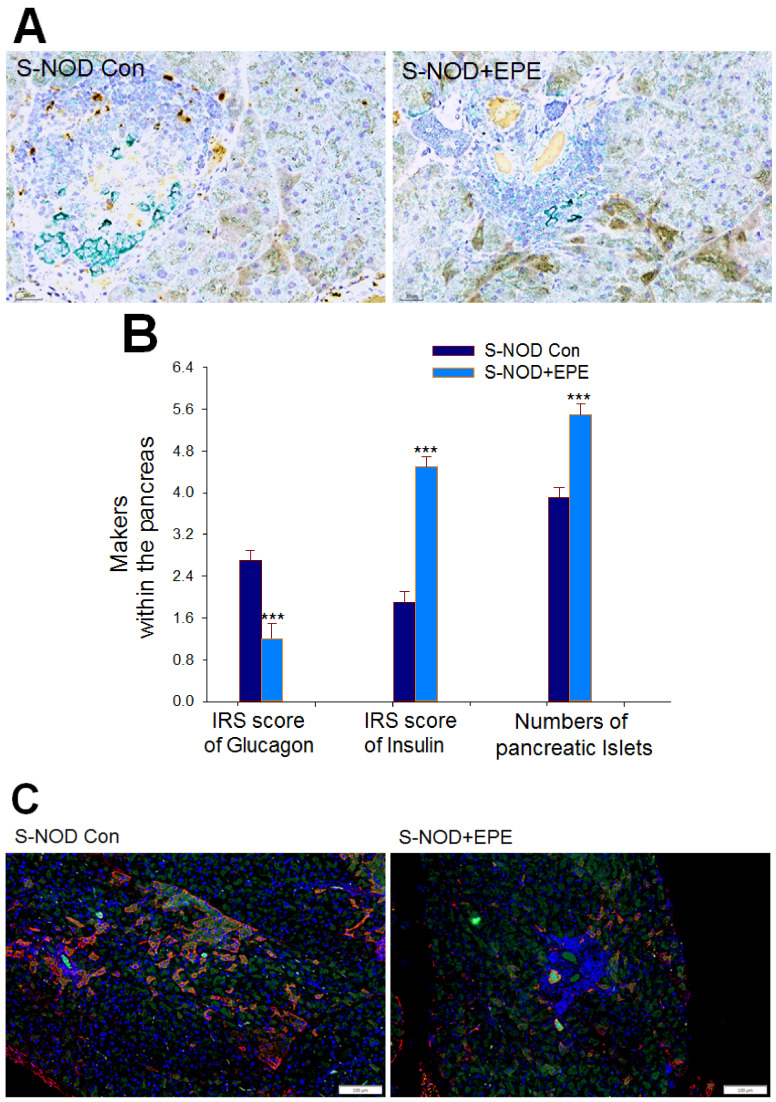
Histological examination of ethyl acetate extract from *Phyllanthus emblica* L. (EPE) on: (**A**) representative pathogenesis photographs of 400× immunohistochemical (IHC) staining of pancreatic insulin-expressing β cells (brown) and glucagon-expressing (green) α cells; (**B**) the immunoreactive system (IRS) score of the pancreas according to both the intensity and quantity of the dyeing signal for makers (including glucagon and insulin), and the numbers of the pancreatic islets within the pancreas; and (**C**–**E**): immunofluorescence staining effects on the expression levels of Bcl-2 (green) and Bax (red) in spontaneous non-obese diabetes (S-NOD) mice (**C**) at magnification of 100×. Left: The expression levels of Bcl-2 and Bax of the Cyp-NOD Con group. Right: The expression levels of Bcl-2 and Bax of the Cyp-NOD +EPE group; (**D**) at magnification of 200×. (**E**) Quantification of effects of EPE on expression levels of Bcl-2 (green) and Bax (red). The percentage (%) of Bcl-2/Bax was calculated using three slides per group under nine visions to count cell numbers using Image J software. The green fluorescence/red fluorescence under one vision was presented as Bcl-2/Bax per cell, and the numbers of Bcl-2/Bax per 100 cells were calculated as % of Bcl-2/Bax. All values are means ± SE (*n* = 7 per group). *** *p* < 0.001 compared to the S-NOD plus vehicle (S-NOD Con) group. EPE: 400 mg/kg body weight.

**Figure 6 ijms-24-09922-f006:**
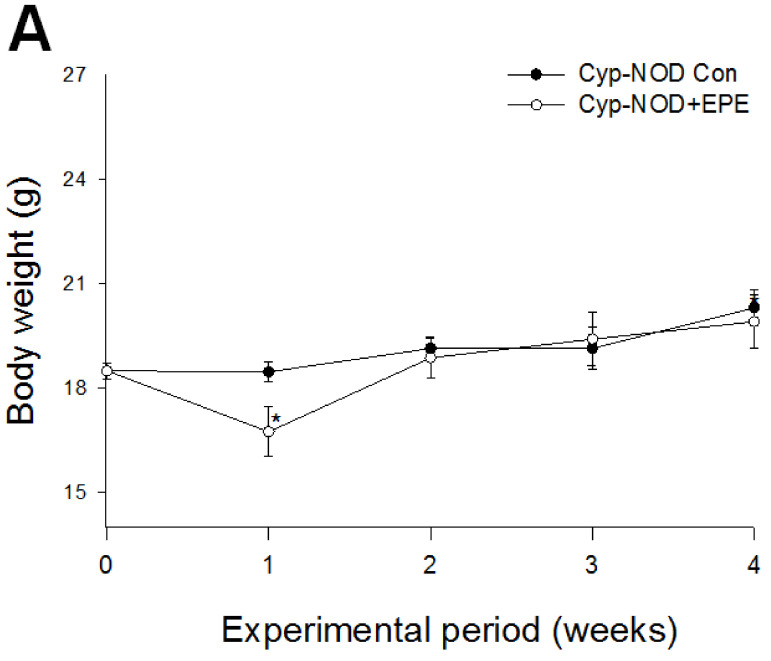
Effects of ethyl acetate from *Phyllanthus emblica* L. (EPE) on (**A**) body weight, (**B**) blood glucose, (**C**) HbA1_C_, (**D**) insulin, (**E**) adiponectin, (**F**) leptin levels, (**G**) blood cytokine levels-1: IL-4, IL-10, and IFN-γ, and (**H**) blood cytokine levels-2: IL-1β, IL-6, and TGF-β1 levels in cyclophosphamide (Cyp)-accelerated non-obese diabetes (Cyp-NOD) mice. All values are means ± SE (*n* = 12). * *p* <0.05 and *** *p* < 0.001 compared with the Cyp induction plus vehicle (distilled water) (Cyp-NOD Con) group. EPE: 400 mg/kg body weight.

**Figure 7 ijms-24-09922-f007:**
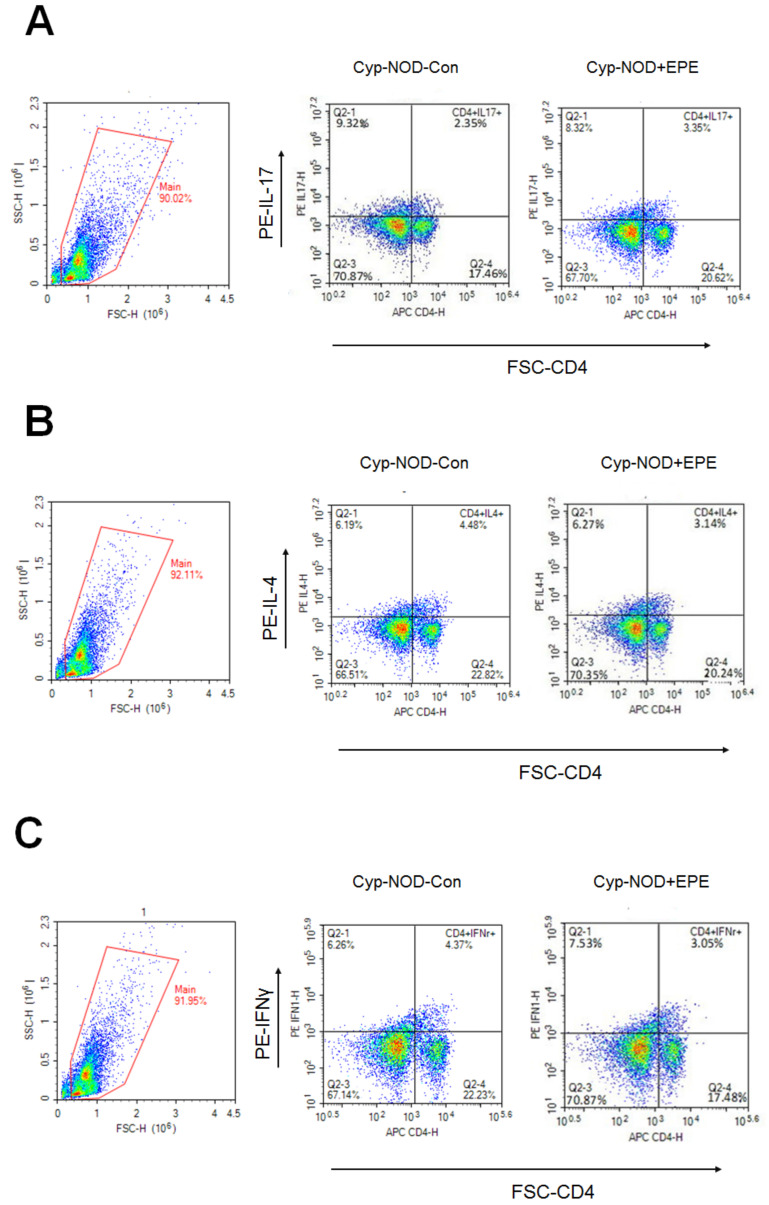
Flow cytometric data of ethyl acetate from *Phyllanthus emblica* L. (EPE) on numbers of (**A**) IL-17, (**B**) IL-4, (**C**) IFNγ, (**D**) Foxp3 subsets of CD4^+^ T cell distribution, and (**E**) four subset CD4^+^ T cell distribution per 10,000 cells (%) in cyclophosphamide (Cyp)-accelerated non-obese diabetes (Cyp-NOD) mice. * *p* < 0.005, ** *p* < 0.001, or *** *p* < 0.0001 compared to the Cyp-NOD mice plus vehicle (distilled water) (Cyp-NOD Con) group. All values are means ± SE (*n* = 12 per group). EPE: 400 mg/kg body weight.

**Figure 8 ijms-24-09922-f008:**
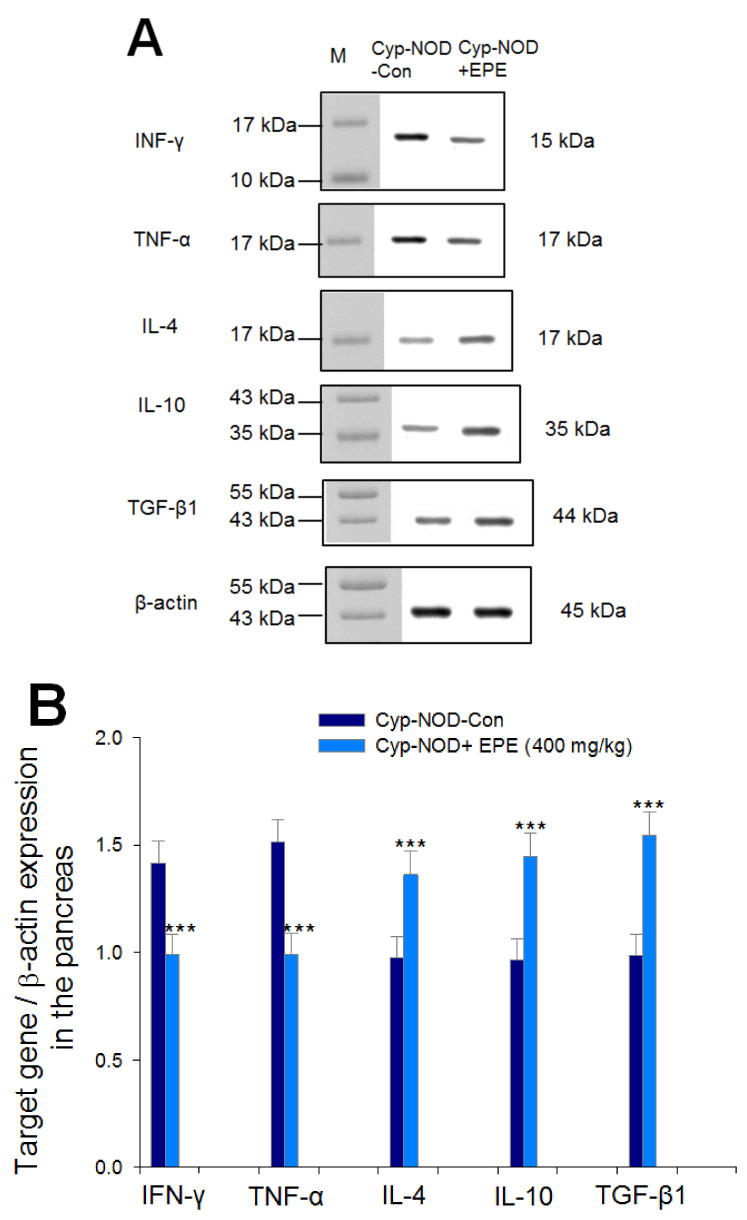
The target gene IFN-γ, TNF-α, IL-4, IL-10, or TGF-β1 expression levels in the pancreas in Cyp-accelerated NOD (Cyp-NOD) mice following treatment with ethyl acetate extract from *Phyllanthus emblica* L. (Cyp-NOD+EPE) (EPE: 400 mg/kg body weight) by Western blotting analysis. (**A**) Representative image. (**B**) Target gene expression quantification to β-actin. Protein was separated by 12% sodium dodecyl sulfate polyacrylamide gel electrophoresis (SDS-PAGE) and detected by Western blotting. *** *p* < 0.0001 compared to the Cyp-NOD mice plus vehicle (distilled water) (Cyp-NOD Con) group. All values are means ± SE (*n* = 12 per group). EPE: 400 mg/kg body weight.

**Figure 9 ijms-24-09922-f009:**
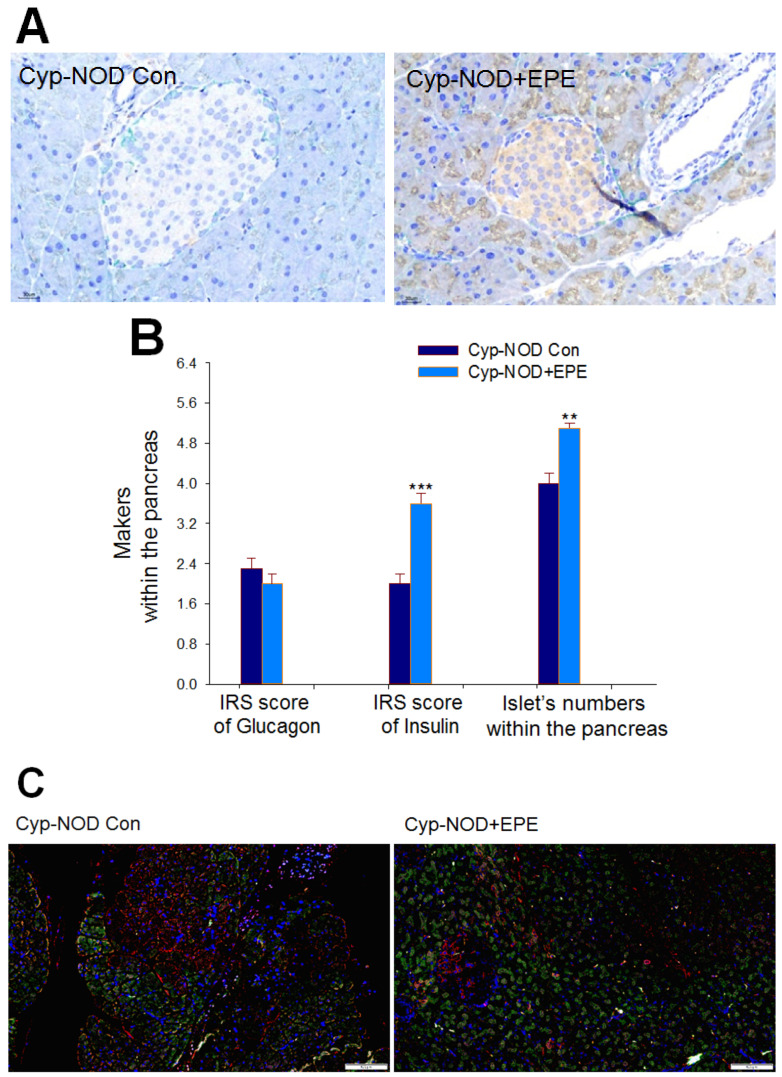
Histological examination of extract of ethyl acetate from *P. emblica* (EPE) on: (**A**) representative pathogenesis photographs of 400× immunohistochemical (IHC) staining of pancreatic insulin-expressing β cells (brown) and glucagon-expressing (green) α cells; (**B**) the immunoreactive system (IRS) score of the pancreas according to both the intensity and quantity of the dyeing signal for makers (including glucagon and insulin) and the numbers of the pancreatic islets within the pancreas; (**C**–**E**): immunofluorescence staining effects on expression levels of Bcl-2 (green) and Bax (red) in cyclophosphamide (Cyp)-accelerated non-obese diabetes (Cyp-NOD) mice (**C**) at magnification of 100x. Left: The expression levels of Bcl-2 and Bax of the Cyp-NOD Con group. Right: The expression levels of Bcl-2 and Bax of the Cyp-NOD+EPE group (**D**) at magnification of 200× and (**E**) quantification of effects of EPE on expression levels of Bcl-2 (green) and Bax (red). The percentage (%) of Bcl-2/Bax was calculated using the method of each group. Three slides under nine visions were observed and the cell numbers were counted by Image J software. The green fluorescence/red fluorescence under one vision was presented as Bcl-2/Bax per cell, and the numbers of Bcl-2/Bax per 100 cells were calculated as % of Bcl-2/Bax. All values are means ± SE (*n* = 12 per group). EPE: 400 mg/kg body weight. ** *p* < 0.01, *** *p* < 0.001.

**Figure 10 ijms-24-09922-f010:**
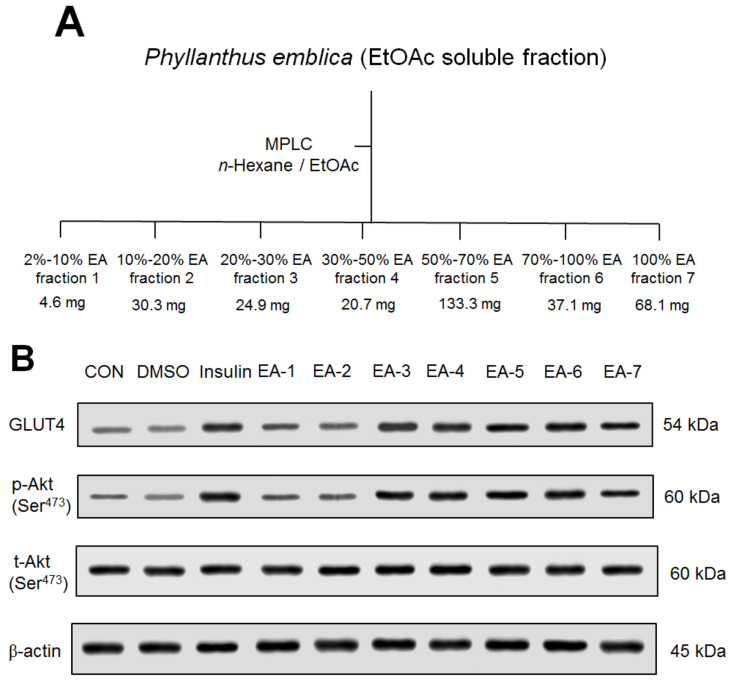
(**A**) Preparation of seven fractions of ethyl acetate extract from *Phyllanthus emblica* L. (EtOAc soluble fractions). The Fruits of *Phyllanthus emblica* were extracted with methanol, and then evaporated and suspended in H_2_O and partitioned with EtOAc. The EtOAc-soluble fraction was then subjected to column chromatography on silica gel using a gradient solvent system of *n*-hexane and EtOAc as a mobile phase and seven fractions were obtained. Seven fractions of EPE are described as EA. EA included 2–10% EA fraction 1 (EA-1), 10–20% EA fraction 2 (EA-2), 20–30% EA fraction 3 (EA-3), 30–50% EA fraction 4 (EA-4), 50–70% EA fraction 5 (EA-5), 70–100% EA fraction 6 (EA-6), and 100% EA fraction 7 (EA-7). (**B**–**D**) Effects of seven fractions of EPE (EA) on expression levels of membrane GLUT4 and phospho-Akt/total-Akt in C2C12 myoblast cells by Western blotting analyses. C2C12 skeletal myoblast cells were treated with seven fractions and equal amounts of lysates were resolved by SDS-PAGE and blotted for GLUT4, total-Akt, and phospho-Akt (Ser^473^). (**B**) Representative blots of seven fractions (EA) in C2C12 myoblast cells. (**C**) Quantification of the expression levels of membrane GLUT4 and (**D**) the ratio of phospho-Akt to total-Akt. All values are means ± S.E. * *p* < 0.05, *** *p* < 0.001 compared with the control group. (**E**) HPLC chromatogram of the polyphenol standards and EA-6 at 280 nm. Determination of phenolic compounds: the peaks indicate the following: 1. Gallic acid (4.3 min); 2. Protocatechuic acid (6.9 min); 3. Chlorogenic acid (9.1 min); 4. Catechin (9.9 min); 5. Protocatechualdehyde (12.9 min); 6. Vanillic acid (13.3 min); 7. Caffeic acid (13.7 min); 8. Epicatechin (14.8 min); 9. Syingic acid (15.5 min); 10. ρ-Coumaric acid (19.1 min); 11. Ferulic acid (21.9 min); 12. Rutin (27.4 min); 13. Narigin (28.3 min); 14. Quercetin (37.2 min); 15. Hesperetin (38.8 min).

**Figure 11 ijms-24-09922-f011:**
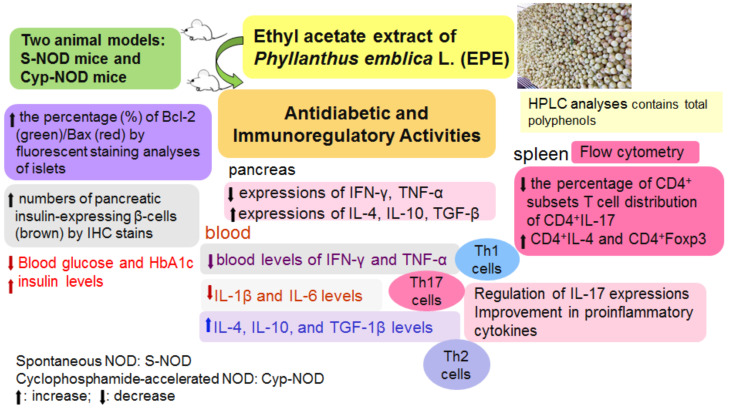
The graphical abstract of the effects of ethyl acetate extract from *Phyllanthus emblica* L. in NOD with spontaneous and cyclophosphamide-accelerated diabetic mice.

**Table 1 ijms-24-09922-t001:** Effects of ethyl acetate from *Phyllanthus emblica* L. (EPE) on relative tissue weight (%) in spontaneous non-obese diabetes (S-NOD) mice.

Parameters	S-NOD Con	S-NOD+EPE
Relative Tissue Weight (%)		
EWAT (%)	1.695 ± 0.878	1.176 ± 0.251
RWAT (%)	0.626 ± 0.091	0.561 ± 0.119
visceral fat (%)	2.321 ± 0.906	1.737 ± 0.325
Liver (%)	5.120 ± 0.289	6.210 ± 0.650
Spleen (%)	0.359 ± 0.013	0.287 ± 0.030 *
Pancreas (%)	0.660 ± 0.046	0.579 ± 0.002
Skeletal muscle (%)	1.484 ± 0.095	1.535 ± 0.103

All values are means ± SE (*n* = 7). * *p* < 0.05, compared to the S-NOD plus vehicle (S-NOD Con) group. EPE: 400 mg/kg body weight.

**Table 2 ijms-24-09922-t002:** Effects of ethyl acetate from *Phyllanthus emblica* L. (EPE) on relative tissue weight (%) in cyclophosphamide (Cyp)-accelerated non-obese diabetes (Cyp-NOD) mice.

Parameters	Cyp-NOD Con	Cyp-NOD+EPE
Relative Tissue Weight (%)		
EWAT (%)	0.682 ± 0.165	0.499 ± 0.163
RWAT (%)	0.518 ± 0.082	0.376 ± 0.083
visceral fat (%)	1.200 ± 0.244	0.876 ± 0.243
Liver (%)	4.934 ± 0.200	4.788 ± 0.082
Spleen (%)	0.879 ± 0.444	0.297 ± 0.024 *
Skeletal muscle (%)	1.772 ± 0.150	1.055 ± 0.134 *

All values are means ± SE (*n* = 12). * *p* < 0.05 compared to the Cyp-NOD mice plus vehicle (Cyp-NOD Con) group. EPE: 400 mg/kg body weight.

## Data Availability

Where no new data were created, or where data is unavailable due to privacy or ethical restrictions. All data used to support the findings of this study are available from the corresponding author, Chun-Ching Shih, upon reasonable request. Corresponding author’s email: ccshih@ctust.edu.tw.

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
