# Peer review of "Antidiabetic and Immunoregulatory Activities of Extract of Phyllanthus emblica L. in NOD with Spontaneous and Cyclophosphamide-Accelerated Diabetic Mice"

_ijms, 2023, doi:10.3390/ijms24129922_

Round 1
Reviewer 1 Report
This paper describes the evaluation of an ethyl acetate extract of Phyllanthus emblica L. (EPE) in a non-obese diabetes (NOD) mice model. According to the results, the extract can cause the lowering of blood glucose, HbA1c, the increasing in insulin and beta cells in the pancreas and a characteristic regulation of some cytokines.
Suggestions:
1. Please rewrite the abstract to make it more concise and understandable.
2. Make sure that all abbreviations have been defined in the manuscript when
they used at first.
3. Please be consistent about how abbreviations appear throughout the document.
4. It would be appropriate to add the initial weights of the aminals in Figure 2.
5. It would be convenient to add the origin of the parameter in the "y" axis title in the graphs that report values in blood, for example: blood glucose levels.
6. Please simplify the y-axis titles in figures 2G, 2H, 3E, 4B, 4C, 6G, 6H, 7E, 8B, 8C, and 9B.
7. There is a missing word (extract) in the title of tables 1, 2, and figures 2, 3, 6, and 7.
8. Perhaps the footer of table 1 can be simplified. Lines 134 and 135.
9. Wrong figure number on line 146.
10. Make the sentence in line 172 more understandable.
11. Wrong word “islet” in figure 5B, 9B on lines 186, 289.
12. Please, indicate how the ISR of figure 5B is obtained and place units (if applicable).
13. There is a repeated phrase in lines 194-196.
14. Improve the sentences of lines 98-200.
15. Please be consistent in how experimental group identifiers appear throughout the document.
16. Maybe "compare" should be "compared" on line 269.
17. It is convenient that the Western blot results have been shown with the molecular weight marker.
18. There are "0" missing in some p-values in figure 2, 7, 8 at lines 129, 263, 278.
19. Be consistent with figure legends on all graphs.
20. Fill in and define "SDS-PAGE". Line 277.
21. An extra space before punctuation marks on lines 297, 339, 341.
22. Words in italics in vitro are required, lines 313 and 624.
23. There may be a typo on line 315.
24. Define EA, Akt and GLUT4. Lines 315 and 316.
25. Please don't miss the units on the axes of the graphs, Figure 10 C and D.
26. The chromatograms in Figure 10 are not identified by words. Lines 324 and 325.
27. There is an extra point on line 342.
28. Rewrite the sentence on line 364 to avoid redundant words.
29. There might be a word missing on lines 416 and 417.
30. There are missing references in the paragraphs of lines 456 and 463.
31. It is necessary to clarify what was the concentration of EPE that was used at 400 mg/kg. Line 521.
32. Missing comma on line 536.
33. Misused verb in line 538.
34. The discussion should be improved by adding why were C2C12 myoblasts used? What about adiponectin and leptin results?
35. Correct word "analysis" in figure 11.

There are some simple grammatical errors throughout the document.
Author Response
Outline the changes made:
Reviewer: 1
- As Reviewer suggests, “1. Please rewrite the abstract to make it more concise and understandable.”
And revised into
---->
Abstract: Oil-Gan, also known as emblica, is the fruit of the genus Phyllanthus emblica L. The fruits are high in nutrients and display excellent health care functions and development values. The primary aim of this study was to investigate the activities of ethyl acetate extract of Phyllanthus emblica L. (EPE) on type 1 diabetes mellitus (T1D) and immunoregulatory activities in non-obese diabetes (NOD) mice with spontaneous and cyclophosphamide (Cyp)-accelerated diabetes. EPE or vehicle-administered to spontaneous NOD (S-NOD) mice or Cyp-accelerated NOD (Cyp-NOD) mice at a dose of 400 mg/kg body weight daily once for 15, or 4 weeks, respectively. At the end, blood samples were collected for biological analyses, and organ tissues were dissected for analyses of histology and immunofluorescence (IF) staining (including expressions of Bcl and Bax), the expression levels of targeted genes by Western blotting, and forkhead box P3 (Foxp3), and helper T lymphocyte 1 (Th1)/Th2/Th17/ Treg regulatory T cell (Treg) cell distribution by flow cytometry. Our results showed that EPE-treated NOD mice or Cyp-accelerated NOD mice display a decrease in levels of blood glucose and HbA1c, but an increase in blood insulin levels. EPE treatment decreased blood levels of IFN-γ and tumor necrosis α (TNF-α) by Th1 cells, and reduced interleukin (IL)-1β and IL-6 by Th17 cells, but increased IL-4, IL-10, and transforming growth factor-β1 (TGF-β1) by Th2 cells in both two mice models by enzyme-linked immunosorbent assay (ELISA) analysis. Flow cytometric data showed that EPE-treated Cyp-NOD mice had decreased the CD4+ subsets T cell distribution of CD4+IL-17 and CD4+ interferon gamma (IFN-γ), but increased the CD4+ subsets T cell distribution of CD4+IL-4 and CD4+Foxp3. Furthermore, EPE-treated Cyp-NOD mice had decreased the percentage per 10000 cells of CD4+IL-17 and CD4+IFNγ, and increased CD4+IL-4 and CD4+Foxp3 compared with the Cyp-NOD Con group (p < 0.001, p < 0.05, p < 0.05, and p < 0.05, respectively). For target gene expression levels in the pancreas, EPE-treated mice had reduced expression levels of inflammatory cytokines, including IFN-γ and TNF-α by Th1 cells, but increased expression levels of IL-4, IL-10, and TGF-1β by Th2 cells in both two mice models. Histological examination of the pancreas revealed that EPE-treated mice had not only increased pancreatic insulin-expressing β cells (brown), and but also enhanced the percentage of Bcl-2 (green) / Bax (red) by IF staining analyses of islets compared with the S-NOD Con and the Cyp-NOD Con mice, implying that EPE displayed the protective effects of pancreas β cells. EPE-treated mice had increased the average immunoreactive system (IRS) score on insulin within the pancreas, and enhanced the numbers of the pancreatic islets. EPE displayed an improvement in the pancreas IRS scores and a decrease in proinflammatory cytokines. Moreover, EPE exerts blood glucose-lowering effects by regulating IL-17 expressions. Collectively, these results implied that EPE inhibits the development of autoimmune diabetes by regulating cytokine expression. Our results demonstrated that EPE had a therapeutic potential in the preventive effects of T1D and immunoregulation as a supplementary.
- As Reviewer suggests, “Make sure that all abbreviations have been defined in the manuscript when they used at first.”
And revised into
----> Yes. We have checked and made sure all abbreviations in the manuscript have been defined when they used at first.
- As Reviewer suggests, “Please be consistent about how abbreviations appear throughout the document.”
And revised into
----> We have revised all abbreviations throughout the manuscript to be consistent.
Abbreviations Used:
CON, control; Cyp, cyclophosphamide; Cyp-NOD, cyclophosphamide-accelerated NOD; EA, seven fractions of ethyl acetate extract of Phyllanthus emblica L.; ELISA, enzyme-linked immunosorbent assay; ESI, electrospray ionization; EPE, ethyl acetate extract of Phyllanthus emblica L.; EWAT, epididymal white adipose tissue; FACS, Fluorescence Activated Cell Sorting; Foxp3, forkhead box P3; HbA1c, glycated hemoglobin; HPLC, high-performance liquid chromatographic; IFN-γ, interferon gamma; IF, Immunofluorescence staining; IL, interleukin; NOD, non-obese diabetes; PDA, photodiode-array; S-NOD, spontaneous non-obese diabetes; TGF-β1, transforming growth factor-β1; Th1, helper T lymphocyte 1; Th2, helper T lymphocyte 2; Treg, regulatory T cell; TNF-α, tumor necrosis α; T1D, Type 1 diabetes mellitus; WAT, white adipose tissue.
- As Reviewer suggests, “It would be appropriate to add the initial weights of the aminals in Figure 2.”
And revised into
----> have revised it,
Figure 2
As Reviewer 2 suggests into line graph:
Figure 6
- As Reviewer suggests, “It would be convenient to add the origin of the parameter in the "y" axis title in the graphs that report values in blood, for example: blood glucose levels.”
And revised into
---->
Figure 2 As Reviewer 2 suggests into line graph:
- As Reviewer suggests, “Please simplify the y-axis titles in figures 2G, 2H, 3E, 4B, 4C, 6G, 6H, 7E, 8B, 8C, and 9B.”
And revised into
----> Figure 2
Figure 3
Figure4
As Reviewer 2 suggests, “ Fig.4B and 4C was combined as thefollowing:
Figure 4
Figure 4. Target gene expression levels (including IFN-γ, TNF-α, IL-4, IL-10, or TGF-β1) in the pancreas following treatment of ethyl acetate extract of Phyllanthus emblica L. (EPE) in spontaneous non-obese diabetes (S-NOD) mice by Western blotting. (A) Representative image; (B) Target gene expression quantification to β-actin.
Figure 6
(H) blood cytokine levels-2: IL-1β, IL-6, and TGF-β1 levels
Figure 7
Figure 8
As Reviewer 2 suggests, “Fig.8B and 8C was combined as thefollowing:
Figure 8. The target gene IFN-γ, TNF-α, IL-4, IL-10, or TGF-β1 expression levels in the pancreas
Figure 9
- As Reviewer suggests, “There is a missing word (extract) in the title of tables 1, 2, and figures 2, 3, 6, and 7.”
And revised into
---->
Table 1. Effects of ethyl acetate of Phyllanthus emblica L. (EPE)
Table 2. Effects of ethyl acetate of Phyllanthus emblica L. (EPE)
Figure 2. Effects of ethyl acetate of Phyllanthus emblica L. (EPE)
Figure 3. Flow cytometric data of ethyl acetate of Phyllanthus emblica L. (EPE)
Figure 6. Effects of ethyl acetate of Phyllanthus emblica L. (EPE)
Figure 7. Flow cytometric data of ethyl acetate of Phyllanthus emblica L. (EPE)
- As Reviewer suggests, “Perhaps the footer of table 1 can be simplified. Lines 134 and 135.”
And revised into
----> All values are means ± SE (n = 7). * p <0.05 compared to the S-NOD plus vehicle (S-NOD Con) group. EPE: 400 mg/kg body weight.
Table 1. Effects of ethyl acetate of Phyllanthus emblica L. (EPE) on relative tissue weight (%) in spontaneous non-obese diabetes (S-NOD) mice.
All values are means ± SE (n = 7). * p <0.05, *** p <0.001 compared to the S-NOD plus vehicle (S-NOD Con) group. EPE: 400 mg/kg body weight.
Table 2. Effects of ethyl acetate of Phyllanthus emblica L. (EPE) on relative tissue weight (%) in cyclophosphamide (Cyp)-accelerated non-obese diabetes (Cyp-NOD) mice.
All values are means ± SE (n = 12). * p <0.05 compared to the Cyp-NOD mice plus vehicle (Cyp-NOD Con) group. EPE: 400 mg/kg body weight.
- As Reviewer suggests, “Wrong figure number on line 146.”
And revised into
----> Flow cytometric data show that EPE-treated S-NOD mice had decreased the CD4+ subsets T cell distribution of CD4+IL-17 (Figure 3A) , but increased the CD4+ subsets T cell distribution of CD4+IL-4 (Figure 3B) and CD4+ forkhead box P3 (Foxp3) (Figure 3D) compared with the S-NOD-Con group. No difference was observed on CD4+ interferon gamma (IFN-γ+) between the S-NOD+EPE group and the S-NOD-Con group (Figure 3C). EPE-treated S-NOD mice had decreased the percentage per 10000 cells of CD4+IL-17, CD4+IL-4, and CD4+Foxp3 compared with the S-NOD-Con group (p < 0.001, p < 0.05, p < 0.05; respectively) (Figure 3E).
- As Reviewer suggests, “Make the sentence in line 172 more understandable.“
And revised into
----> The green fluorescence / red fluorescence represents Bcl-2 / Bax per cell under one vision. EPE-treated S-NOD mice had significantly increased the percentage of Bcl-2 (green) / Bax (red) compared with the S-NOD-Con mice by fluorescent staining analysis of islets counting by Image J software at 100x or 200x (p < 0.001, p < 0.001, respectively) (Figure 5C,D,E).
- As Reviewer suggests, “Wrong word “islet” in figure 5B, 9B on lines 186, 289.“
And revised into
---->
2.1.1.5. ….. EPE- treated S-NOD mice had significantly decreased the average immunoreactive system (IRS) score of glucagon but increased the average IRS score of insulin within the pancreas, and enhanced the number of the pancreatic islets compared with the S-NOD Con group (p < 0.001, p < 0.001, p < 0.001, respectively) (Figure 5B). Administration of EPE to S-NOD mice had increased pancreatic insulin-expressing β cells (brown) compared with S-NOD Con mice (p < 0.001) (Figure 5B).
2.1.2.5. … EPE-treated Cyp-NOD mice had significantly increased the average IRS score of insulin within the pancreas, and enhanced the number of the pancreatic islets compared with the Cyp-NOD Con group (p < 0.001, p < 0.01, respectively) (Figure 9B).
The figures 5B and 9B have revised into the following:
Figure 5B
Figure 5. Histological examination of extract of ethyl acetate of Phyllanthus emblica L. (EPE) on: (A) Representative pathogenesis photographs of 400× immunohistochemical (IHC) staining of pancreatic insulin-expressing β cells (brown) and glucagon-expressing (green) α cells; (B) the immunoreactive system (IRS) score of pancreas according to both the intensity and quantity of the dyeing signal for makers (including glucagon and insulin), and the numbers of the pancreatic islets within the pancreas; (B) Representative pathogenesis photographs of 400× immunohistochemical (IHC) staining of pancreatic insulin-expressing β cells (brown) and glucagon-expressing (green) α cells;
Figure 9B
Figure 9 (B) the immunoreactive system (IRS) score of pancreas according to both the intensity and quantity of the dyeing signal for makers (including glucagon and insulin), and the numbers of the pancreatic islets within the pancreas;
- As Reviewer suggests, “Please, indicate how the ISR of figure 5B is obtained and place units (if applicable).”
And revised into
---->
immunoreactive system (IRS) score according to both the intensity and quantity of the dyeing signal for makers (including glucagon and insulin) within the pancreas using the following criteria of IRS score: 0-1: negative; 2-3: mild positive; 4-8: moderate positive; 9-12 strongly positive [57,58]. A (quantity) included the following issues: 0 = no positive cells, 1 = <10% of positive cells, 2 = 10 – 50% positive cells, 3 = 51 – 80% positive cells, and 4 = >80% positive cells. B (intensity) included the following conditions: 0 = no color, 1 = mild positive, 2 = moderate positive, and 3 = strong positive. IRS score, multiplication of A and B. Final IRS score = A (quantity)×B (intensity) = 0-12 [58]. Futhermore, the number of the pancreatic islet were also calculated within the pancreas by pathology veterinarian.
No units
Figure 5B
Figure 9B
- As Reviewer suggests, “There is a repeated phrase in lines 194-196.”
And revised into
----> have deleted it
- As Reviewer suggests, “Improve the sentences of lines 98-200.”
And revised into
----> Figure 4. Effects of ethyl acetate extract of Phyllanthus emblica L. (EPE) on expression levels of target genes (including IFN-γ, TNF-α, IL-4, IL-10, or TGF-β1) within the pancreas of spontaneous non-obese diabetes (S-NOD) mice by Western blotting. (A) Representative image; (B) (C) Target gene expression quantification to β-actin. Protein was separated by 12% sodium dodecyl sulfate polyacrylamide gel electrophoresis (SDS-PAGE) detected by Western blotting. *** p < 0.001 compared to the S-NOD plus vehicle (distilled water) (S-NOD Con) group. All values are means ± SE (n = 7 per group). EPE: 400 mg/kg/ body weight.
The percentage (%) of Bcl-2/Bax was calculated by three slides per group under nine visions to count cell numbers using Image J software.
- As Reviewer suggests, “Please be consistent in how experimental group identifiers appear throughout the document.”
And revised into
----> Yes
- As Reviewer suggests, “Maybe "compare" should be "compared" on line 269.”
And revised into
----> compared
- As Reviewer suggests, “It is convenient that the Western blot results have been shown with the molecular weight marker.”
And revised into
----> have revised it
Figure 4
Figure 8
- As Reviewer suggests, “There are "0" missing in some p-values in figure 2, 7, 8 at lines 129, 263, 278.”
And revised into
----> p-values are very small but still exist in Figure 2, 7, 8.
- As Reviewer suggests, “Be consistent with figure legends on all graphs.”
And revised into
----> All figure legends were consistent with all graphs
- As Reviewer suggests, “Fill in and define "SDS-PAGE". Line 277.”
And revised into
----> Protein was separated by 12% sodium dodecyl sulfate polyacrylamide gel electrophoresis (SDS-PAGE) detected by Western blotting.
- As Reviewer suggests, “An extra space before punctuation marks on lines 297, 339, 341.”
And revised into
----> have revised it
- As Reviewer suggests, “Words in italics in vitro are required, lines 313 and 624.”
And revised into
----> in vitro
- As Reviewer suggests, “There may be a typo on line 315.”
And revised into
----> have revised it
- As Reviewer suggests, “Define EA, Akt and GLUT4. Lines 315 and 316.”
And revised into
---->
2.2.1. Seven fractions of EPE on Targeted Gene Expressions in vitro
The preparation of seven fractions of EPE procedure was described as Figure 10A. Seven fractions of EPE (EA) are described as EA. EA included 2%~10% EA fraction 1 (EA-1), 10%~20% EA fraction (EA-2), 20%~30% EA fraction 3 (EA-3), 30%~50% EA fraction 4 (EA-4), 50%~70% EA fraction 5 (EA-5), 70%~100% EA fraction 6 (EA-6), and 100% EA fraction 7 (EA-7). Glucose transporter 4 (GLUT4) is known to play a central role in blood glucose homeostasis [26]. Either stimulation of insulin or contraction facilitates glucose uptake by translocate GLUT4 to the cell membrane [27,28]. The mechanisms of promoting glucose uptake into skeletal muscle included the insulin-dependent mechanisms leading to activation of Akt/PKB and contraction-regulated stimulation [29,30]. Therefore, we choose membrane GLUT4 and activation of Akt/PKB as anti-diabetic target genes. Figure 10B–10C show the insulin-, EA-3-, EA-4-, EA-5-, EA-6-, and EA-7-treated groups had increased expression levels of membrane GLUT4 or raised Akt activation (phospho-Akt/ total-Akt; p-Akt/ t-Akt) in comparison to the control group. Our findings have shown that EA-6 displays the best activity of phospho-Akt/ total-Akt and the expression levels of GLUT4 in vitro, and then the study was designed to explore the marker ingredient of EPE responsible for the anti-diabetes.
- As Reviewer suggests, “Please don't miss the units on the axes of the graphs, Figure 10 C and D.”
And revised into
---->
Figure 10
- As Reviewer suggests, “The chromatograms in Figure 10 are not identified by words. Lines 324 and 325.”
And revised into
---->(E) HPLC chromatogram
Figure 10. (E) HPLC chromatogram of the polyphenol standards and EA-6 at 280 nm. Determination of phenolic compounds: the peaks indicate the following: 1. Gallic acid (4.3 min); 2. Protocatechuic acid (6.9 min); 3. Chlorogenic acid (9.1 min); 4. Catechin (9.9 min); 5. Protocatechualdehyde (12.9 min); 6. Vanillic acid (13.3 min); 7. Caffeic acid (13.7 min); 8. Epicatechin (14.8. min); 9. Syingic acid (15.5 min); 10. ρ-Coumaric acid (19.1 min); 11. Ferulic acid (21.9 min); 12. Rutin (27.4 min); 13. Narigin (28.3min); 14. Quercetin (37.2 min); 15. Hesperetin (38.8 min).
2.2.2. Analysis of EA-6 Fraction
Polyphenolic compounds are natural antioxidants that play major functional role in plants by scavenging free radicals [31]. Polyphenolic compounds can be divided into two large categories: flavonoids and phenolic acids.
This study was designed to examine whether polyphenolic compounds found in EA-6 fraction or not. The retention time of gallic acid was 4.3 minute as reference compound to analyze the EA-6 fraction qualitatively and quantitatively. As shown in Figure 10E, the major absorption peaks of phenolic acids were consistent with the Photo Diode Array (PDA) spectra of the reference materials.
As shown in Figure 10E, the main ingredient phenolic compound is gallic acid found in EA-6 (33.4%).
- As Reviewer suggests, “There is an extra point on line 342.”
And revised into
----> have deleted it
- As Reviewer suggests, “Rewrite the sentence on line 364 to avoid redundant words.”
And revised into
----> The mobile phase contained Acetonitrile (solvent A) and acidified water with trifluoroacetic acid (0.05%, solvent B).
4.3. Preparation of Ethyl Acetate Extract of Phyllanthus emblica L. (EPE)
The 7.3 kg fruits of Phyllanthus emblica L. (Figure 1A) were dried and pulverized into 1.34 kg fine powder. The fine powder was extracted with 6.3 L methanol three times at room temperature to collect the supernatant to concentration under vacuum, and then the crude methanolic extract (331.86 g) was obtained. The crude methanolic extract was subjected to three times of suspension in H2O and partition with EtOAc, respectively, and followed by concentration under reduced pressure, and then the H2O fraction (323.71 g) and the EtOAc fraction (47.28 g) was obtained (Figure 1B). The EtOAc fraction was employed for animal study.
- As Reviewer suggests, “There might be a word missing on lines 416 and 417.”
And revised into
----> have revised it
- As Reviewer suggests, “There are missing references in the paragraphs of lines 456 and 463.”
And revised into
----> have revised it
- As Reviewer suggests, “It is necessary to clarify what was the concentration of EPE that was used at 400 mg/kg. Line 521.”
And revised into
----> EPE: 400 mg/kg body weight
- As Reviewer suggests, “Missing comma on line 536.”
And revised into
----> have added it
- As Reviewer suggests, “Misused verb in line 538.”
And revised into
----> have deleted it
- As Reviewer suggests, “The discussion should be improved by adding why were C2C12 myoblasts used? What about adiponectin and leptin results?”
And revised into
---->
Discussion
Our findings showing that administration of EPE to S-NOD and Cyp-NOD mice had increased blood levels of adiponectin and leptin. Wu et al. demonstrated that treatment with globular domain of adiponectin increased in glucose uptake [47], suggesting EPE could regulate secretion of adiponectin to lead to an increase in glucose uptake, and in turn to control glucose homeostasis. Evidence has shown that leptin could substitute for insulin to control blood sugar fluctuations in patients with type 1 diabetes [48]. Nevertheless, the mechanism of action still remains unknown. This study demonstrated that EPE had favor effects on leptin levels, implying that EPE playing a critical role in glucose metabolism partly by enhancement of blood leptin levels.
Skeletal muscle is the major tissue responsible for insulin-mediated glucose utilization [49]. Due to its expression of GLUT4 representative to human skeletal muscle cells, the C2C12 myoblast cell line is employed in vitro study. Since membrane GLUT4 and p-Akt/ t-Akt play the critical roles in Type 1 diabetes, the cell culture analyses of expressions of GLUT4 and p-Akt/ t-Akt were determined to clarify the anti-diabetic activity using Western blotting. Our finding showed that EA-6 displayed the best activity of seven fractions on increased expression levels of membrane GLUT4 and phospho-Akt/ total-Akt, and then we try to find the marker ingredient using HPLC analysis.
- As Reviewer suggests, “Correct word "analysis" in figure 11.”
And revised into
----> Figure 11

Reviewer 2 Report
The manuscript by Lin et al, describes the antidiabetic and Immunoregulatory activities of extract of Phyllanthus emblica L. in NOD with Spontaneous and Cyclo-phosphamide-accelerated diabetic mice. The question posed by the authors is well defined; the methods are appropriate and well described; the data are sound; the manuscript adheres to the relevant standards for reporting and data deposition; the discussion and conclusions are well balanced and adequately supported by the data. However few technical questions can be raised.
The authors studied the effects of only one dose of ethyl acetate extract of Phyllanthus emblica L. (EPE) in Spontaneous Non-obese Diabetes (S-NOD) T1DM Mice. 2. The authors failed to demonstrate dose dependent effect of acacetin since they used only one dose throughout the manuscript.
NOD mice are used as an animal model for type 1 diabetes. Diabetes develops in NOD mice as a result of insulitis, a leukocytic infiltrate of the pancreatic islets. The onset of diabetes is associated with a moderate glycosuria and a non-fasting hyperglycemia. The treatment with EPE for 15 weeks, the S-NOD+EPE mice had decreased body weight and no change in the relative tissue weight compared with the S-NOD-Con group. What could be the reason?
What is the rationale of using the crude methanolic extract suspended and partitioned with H2O and EtOAc (1:1) ?
It would be better represented as a line graph showing week-wise progression in blood glucose and changes in body weight instead of initial and final readings.
Fig. 2B & Fig. 6B: Significance is mentioned within the bar. The author has to correct correct such errors.
Results section should be elaborated. For example, section 2.2.1 is not clear. The authors need to elaborate the changes observed in each group.
All subheadings in the result section have to be revised. Please give an appropriate title based on your significant findings. Please don’t give the titles as western blot/immunostaining.
Fig 6H represents levels of TGF-b1, Fig 8A represents western blotting of TGF-b1, whereas Fig 8C represents whole TGF-b. Why?
Fig.8B and 8C can be combined
Looks good. Minor grammatical errors can be fixed
Author Response
Outline the changes made:
Reviewer: 2
- As Reviewer suggests, “The authors failed to demonstrate dose dependent effect of acacetin since they used only one dose throughout the manuscript.”
And revised into
----> Because the expensive cost and difficulty acquisition of NOD mice, we employed only one dose in the animal study. Therefore, our present results could not ascertain dose dependent effect of EPE.
- As Reviewer suggests, “NOD mice are used as an animal model for type 1 diabetes. Diabetes develops in NOD mice as a result of insulitis, a leukocytic infiltrate of the pancreatic islets. The onset of diabetes is associated with a moderate glycosuria and a non-fasting hyperglycemia. The treatment with EPE for 15 weeks, the S-NOD+EPE mice had decreased body weight and no change in the relative tissue weight compared with the S-NOD-Con group. What could be the reason?”
Ans: (We made mistakes on writing down the data of S-NOD Con group and the S-NOD +EPE group from SPSS software). S-NOD+EPE mice decreased relative spleen weights compared with the S-NOD-Con group (p <0.05, respectively) (Table 1). Since the big variation between two groups (in addition to the relative spleen weights), there is no change in the relative tissue weight compared with the S-NOD-Con group. These results may imply that the relative weights of EWAT and visceral fat could not as an indicator of T1D in both two mice models.
And revised into
---->
Table 1. Effects of ethyl acetate of Phyllanthus emblica L. (EPE) on relative tissue weight (%) in spontaneous non-obese diabetes (S-NOD) mice.
|
parameters |
S-NOD Con |
S-NOD +EPE |
|
Relative tissue weight (%) |
|
|
|
EWAT (%) |
1.695 ± 0.878 |
1.176 ± 0.251 |
|
RWAT (%) |
0.626 ± 0.091 |
0.561 ± 0.119 |
|
visceral fat (%) |
2.321 ± 0.906 |
1.737 ± 0.325 |
|
Liver (%) |
5.120 ± 0.289 |
6.210 ± 0.650 |
|
Spleen (%) |
0.359 ± 0.013 |
0.287 ± 0.030* |
|
Pancreas (%) |
0.660 ± 0.046 |
0.579 ± 0.002 |
|
Skeletal muscle (%) |
1.484 ± 0.095 |
1.535 ± 0.103 |
All values are means ± SE (n = 7). * p <0.05, *** p <0.001 compared to the S-NOD plus vehicle (S-NOD Con) group. EPE: 400 mg/kg body weight.
Table 2. Effects of ethyl acetate of Phyllanthus emblica L. (EPE) on relative tissue weight (%) in cyclophosphamide (Cyp)-accelerated non-obese diabetes (Cyp-NOD) mice.
Table 2. Effects of ethyl acetate of Phyllanthus emblica L. (EPE) on relative tissue weight (%) in cyclophosphamide (Cyp)-accelerated non-obese diabetes (Cyp-NOD) mice.
|
parameters |
Cyp-NOD Con |
Cyp-NOD +EPE |
|
Relative tissue weight (%) |
|
|
|
EWAT (%) |
0.682 ± 0.165 |
0.499 ± 0.163 |
|
RWAT (%) |
0.518 ± 0.082 |
0.376 ± 0.083 |
|
visceral fat (%) |
1.200 ± 0.244 |
0.876 ± 0.243 |
|
Liver (%) |
4.934 ± 0.200 |
4.788 ± 0.082 |
|
Spleen (%) |
0.879 ± 0.444 |
0.297 ± 0.024* |
|
Skeletal muscle (%) |
1.772 ± 0.150 |
1.055 ± 0.134* |
All values are means ± SE (n = 12). * p <0.05 compared to the Cyp-NOD mice plus vehicle (Cyp-NOD Con) group. EPE: 400 mg/kg body weight.
- As Reviewer suggests, “What is the rationale of using the crude methanolic extract suspended and partitioned with H2O and EtOAc (1:1) ?”
And revised into
----> Have deleted it (1:1)
4.3. Preparation of Ethyl Acetate Extract of Phyllanthus emblica L. (EPE)
The 7.3 kg fruits of Phyllanthus emblica L. (Figure 1A) were dried and pulverized into 1.34 kg fine powder. The fine powder was extracted with 6.3 L methanol three times at room temperature to concentration under vacuum, and then the crude methanolic extract (331.86 g) was obtained. The crude methanolic extract was subjected to three times of suspension in H2O and partition with EtOAc, respectively, and followed by concentration under reduced pressure, and then the H2O fraction (323.71 g) and the EtOAc fraction (47.28 g) was obtained (Figure 1B). The EtOAc fraction was employed for animal study.
- As Reviewer suggests, “It would be better represented as a line graph showing week-wise progression in blood glucose and changes in body weight instead of initial and final readings.”
And revised into
----> Because it is difficult to acquire and breed NOD mice, we are afraid to make them die and not to collect blood weekly. When mice gradually grow up, we collect blood to analyze blood glucose levels. Especially in the part II’s Cyp-NOD mice, mice seem to be not so healthy due to the chemical cyclophosphamide, and therefore we carefully only collected blood to test blood glucose levels in the initial and the end of experiment.
Figure 2 A
Figure 2
Figure 6
Figure 6
- As Reviewer suggests, “Fig. 2B & Fig. 6B: Significance is mentioned within the bar. The author has to correct correct such errors.”
And revised into
---->
2.1. Animal Study
2.1.1. Part I: Effects of Ethyl Acetate Extract of Phyllanthus emblica L. (EPE) in S-NOD T1D Mice
2.1.1.1. Body Weight, Relative Tissue Weight, Blood Glucose, HbA1C, Insulin, Adiponectin, and Leptin Levels
As shown in Figure 2A, treatment with EPE for 15 weeks, the S-NOD+EPE mice had decreased body weight compared with the S-NOD-Con group treatment with EPE from week 4 to week 15 (Figure 2A). S-NOD+EPE mice decreased relative EWAT, visceral fat, and spleen weights compared with the S-NOD-Con group (p <0.001, p <0.05, p <0.05, respectively) (Table 1). Figure 2B showed blood glucose levels of the initial time and the EPE-treated mice during the experimental 15 weeks. S-NOD+EPE mice had lowered blood glucose levels compared with the S-NOD-Con group at week 11 and week 15 (p < 0.05, p < 0.001, respectively) (Figure 2B). S-NOD+EPE mice had both lowered blood HbA1C concentrations compared with the S-NOD-Con group (p < 0.001) (Figure 2C).
2.1.2. Part II: Effects of EPE in Cyp-Induced T1D Mice
2.1.2.1. Body Weight, Relative Tissue Weight, Blood Glucose, HbA1C, Insulin, Adiponectin, and Leptin Levels
As shown in Figure 6, treatment with EPE for 4 weeks, EPE-treated mice had decreased body weight at week 1 (p < 0.05), and there is no difference in week 2, 3, and the final body weight (week 4) between the Cyp-NOD+EPE group and the Cyp-NOD-Con group (Figure 6A). Cyp-NOD+EPE mice had both decreased relative tissue weights of EWAT, RWAT, visceral fat, spleen, and skeletal muscle compared with the Cyp-NOD-Con group (p < 0.05, p < 0.05, p < 0.05, p < 0.05, p < 0.05, respectively) (Table 2). As shown in Figure 6B, there is no difference in blood glucose levels between Cyp-NOD Con mice and Cyp-NOD+EPE mice at the initial time (week 0). At the final time (week 4), Cyp-NOD+EPE mice had significantly lowered blood glucose levels compared with the Cyp-NOD Con group (p < 0.001) (Figure 6B). Cyp-NOD+EPE mice had significantly lowered blood HbA1C concentrations compared with the Cyp-NOD Con group (p < 0.001) (Figure 6C).
- As Reviewer suggests, “Results section should be elaborated. For example, section 2.2.1 is not clear. The authors need to elaborate the changes observed in each group.”
And revised into
---->
2.2. Preparation of Seven Fractions of EPE
2.2.1. Seven Fractions of EPE on Targeted Gene Expressions in vitro
The preparation of seven fractions of EPE procedure was described as Figure 10A. Seven fractions of EPE (EA) are described as EA. EA included 2%~10% EA fraction 1 (EA-1), 10%~20% EA fraction (EA-2), 20%~30% EA fraction 3 (EA-3), 30%~50% EA fraction 4 (EA-4), 50%~70% EA fraction 5 (EA-5), 70%~100% EA fraction 6 (EA-6), and 100% EA fraction 7 (EA-7). Glucose transporter 4 (GLUT4) is known to play a central role in blood glucose homeostasis [26]. Either stimulation of insulin or contraction facilitates glucose uptake by translocate GLUT4 to the cell membrane [27,28]. The mechanisms of promoting glucose uptake into skeletal muscle included the insulin-dependent mechanisms leading to activation of Akt/PKB and contraction-regulated stimulation [29,30]. Therefore, we choose membrane GLUT4 and activation of Akt/PKB as anti-diabetic target genes. Figure 10B–10C show the insulin-, EA-3-, EA-4-, EA-5-, EA-6-, and EA-7-treated groups had increased expression levels of membrane GLUT4 or raised Akt activation (phospho-Akt/ total-Akt; p-Akt/ t-Akt) in comparison to the control group. Our findings have shown that EA-6 displays the best activity of phospho-Akt/ total-Akt and the expression levels of GLUT4 in vitro, and then the study was designed to explore the marker ingredient of EPE responsible for the anti-diabetes.
- As Reviewer suggests, “All subheadings in the result section have to be revised. Please give an appropriate title based on your significant findings. Please don’t give the titles as western blot/immunostaining.”
And revised into
---->
2.1.1.4. Western Blotting of Interferon Gamma (IFN-γ) and Tumor Necrosis α (TNF-α), IL-4, IL-10, and Transforming Growth Factor-β1 (TGF-β1) Expressions
Target Genes Expression Levels in The Pancreas
2.1.1.5. Immunohistochemical (IHC) Staining
Insulin-Expressing β cells, IRS Score on Glucagon and Insulin, and the Pancreatic Islet Numbers
2.1.1.6. Immunofluorescence Staining (IF)
The Percentage of Bcl-2 / Bax
2.1.2.4. Western Blotting of IFN-γ and TNF-α, IL-4, IL-10, and TGF-β1 Expression Levels
2.1.2.4. Target Gene Expression Levels in The Pancreas
2.1.2.5. Immunohistochemical (IHC) Staining
Insulin-Expressing β cells, IRS Score on Insulin, and the Pancreatic Islet Numbers
2.1.2.6. Immunofluorescence (IF) Staining
The Percentage of Bcl-2 / Bax
- As Reviewer suggests, “Fig 6H represents levels of TGF-b1, Fig 8A represents western blotting of TGF-b1, whereas Fig 8C represents whole TGF-b. Why?” “Fig.8B and 8C can be combined.”
And revised into
---->
Figure 6
Figure 6. (H) blood cytokine levels-2: IL-1β, IL-6, and TGF-β1 levels
Fig.8B and 8C was combined as thefollowing:
Figure 8. The target gene IFN-γ, TNF-α, IL-4, IL-10, or TGF-β1 expression levels in the pancreas
Fig.4B and 4C was combined as thefollowing:
Figure 4
Figure 4. Target gene expression levels (including IFN-γ, TNF-α, IL-4, IL-10, or TGF-β1) in the pancreas following treatment of ethyl acetate extract of Phyllanthus emblica L. (EPE) in spontaneous non-obese diabetes (S-NOD) mice by Western blotting. (A) Representative image; (B) Target gene expression quantification to β-actin.

Reviewer 3 Report
Lin et al in the present report show that extract of Phyllanthus embolic L. prevents the development of hyperglycaemia in NOD mice.
Although authors have enough data to support their hypothesis I have following suggestions:
1. Consistency in writing is not good as authors write few places Type 1 diabetes and few places T1D same go for a number of abbreviations i.e. Th1, Th17 and so on.
2. Authors write TNF-beta, my understanding is that they meant TGF-beta if so please correct typing errors.
3. Figures could be restructured for example instead of putting all the sub figures in vertical arrangement (A-H) authors could got for both horizontal (A-C) in the next line (D-F) and vertical.
4. Regulatory T cells also produce IL-10 and TGF-beta but authors mainly discussing Th2 cells.
5. Abstract is too long.
6. In abstract line number 27 authors write insulin was decreased (I think it is typo).
7. Animal ethical consideration part is missing.
8. Conclusion part could be bit shorten.
9. In methods part it is not clear the age of mice when they first received the treatment.
10 Figure 1 is not mentioned or referred in the text anywhere in the manuscript.
Language should be improved.
Author Response
/c/v
Outline the changes made:
Reviewer: 3
- As Reviewer suggests, “Consistency in writing is not good as authors write few places Type 1 diabetes and few places T1D same go for a number of abbreviations i.e. Th1, Th17 and so on.’
And revised into
----> We have revised all abbreviations throughout the manuscript to be consistent.
Abbreviations Used:
CON, control; Cyp, cyclophosphamide; Cyp-NOD, cyclophosphamide-accelerated NOD; EA, seven fractions of EPE (including EA-1, EA-2, EA-3, EA-A, EA-5, EA-6, and EA-7); ELISA, enzyme-linked immunosorbent assay; ESI, electrospray ionization; EPE, ethyl acetate extract of Phyllanthus emblica L.; EWAT, epididymal white adipose tissue; FACS, Fluorescence Activated Cell Sorting; Foxp3, forkhead box P3; HbA1c, glycated hemoglobin; HPLC, high-performance liquid chromatographic; IFN-γ, interferon gamma; IF, Immunofluorescence staining;
IL, interleukin; NOD, non-obese diabetes; PDA, photodiode-array; S-NOD, spontaneous non-obese diabetes; TGF-β1, transforming growth factor-β1; Th1, helper T lymphocyte 1; Th2, helper T lymphocyte 2; Treg, regulatory T cell; TNF-α, tumor necrosis α; T1D, Type 1 diabetes mellitus; WAT, white adipose tissue.
- As Reviewer suggests, “Authors write TNF-beta, my understanding is that they meant TGF-beta if so please correct typing errors.”
And revised into
----> We have revised all TNF-beta into TGF-β1 in the manuscript and Figs.
- As Reviewer suggests, “Figures could be restructured for example instead of putting all the sub figures in vertical arrangement (A-H) authors could got for both horizontal (A-C) in the next line (D-F) and vertical.”
Ans:
Thank you. Because other Reviewer suggests Figs of body weight into line plots, we think the present suggestion may be not equal size if we combine into A-C as the following:
and therefore we think let the assistant editor to help the arrangement of plots
- As Reviewer suggests, “Regulatory T cells also produce IL-10 and TGF-beta but authors mainly discussing Th2 cells.”
And revised into
----> Discussion
Our results in part I’ s S-NOD mice showed that EPE-treated S-NOD mice had decreased the pancreatic expression levels of IFN-γ and TNF-α to lead to reduced blood levels of IFN-γ and TNF-α cytokines, which are associated with Th1 cell cytokines, but increased the expression levels of IL-4, IL-10, and TGF-β1 (with relation to Th2 cell cytokines) to lead to enhancement of blood levels of IL-4, IL-10, and TGF-β1 cytokines compared with S-NOD Con mice.
- As Reviewer suggests, “Abstract is too long.”
And revised into
----> Abstract: Oil-Gan, also known as emblica, is the fruit of the genus Phyllanthus emblica L. The fruits are high in nutrients and display excellent health care functions and development values. The primary aim of this study was to investigate the activities of ethyl acetate extract of Phyllanthus emblica L. (EPE) on type 1 diabetes mellitus (T1D) and immunoregulatory activities in non-obese diabetes (NOD) mice with spontaneous and cyclophosphamide (Cyp)-accelerated diabetes. EPE or vehicle-administered to spontaneous NOD (S-NOD) mice or Cyp-accelerated NOD (Cyp-NOD) mice at a dose of 400 mg/kg body weight daily once for 15, or 4 weeks, respectively. At the end, blood samples were collected for biological analyses, and organ tissues were dissected for analyses of histology and immunofluorescence (IF) staining (including expressions of Bcl and Bax), the expression levels of targeted genes by Western blotting, and forkhead box P3 (Foxp3), and helper T lymphocyte 1 (Th1)/Th2/Th17/ Treg regulatory T cell (Treg) cell distribution by flow cytometry. Our results showed that EPE-treated NOD mice or Cyp-accelerated NOD mice display a decrease in levels of blood glucose and HbA1c, but an increase in blood insulin levels. EPE treatment decreased blood levels of IFN-γ and tumor necrosis α (TNF-α) by Th1 cells, and reduced interleukin (IL)-1β and IL-6 by Th17 cells, but increased IL-4, IL-10, and transforming growth factor-β1 (TGF-β1) by Th2 cells in both two mice models by enzyme-linked immunosorbent assay (ELISA) analysis. Flow cytometric data showed that EPE-treated Cyp-NOD mice had decreased the CD4+ subsets T cell distribution of CD4+IL-17 and CD4+ interferon gamma (IFN-γ), but increased the CD4+ subsets T cell distribution of CD4+IL-4 and CD4+Foxp3. Furthermore, EPE-treated Cyp-NOD mice had decreased the percentage per 10000 cells of CD4+IL-17 and CD4+IFNγ, and increased CD4+IL-4 and CD4+Foxp3 compared with the Cyp-NOD Con group (p < 0.001, p < 0.05, p < 0.05, and p < 0.05, respectively). For target gene expression levels in the pancreas, EPE-treated mice had reduced expression levels of inflammatory cytokines, including IFN-γ and TNF-α by Th1 cells, but increased expression levels of IL-4, IL-10, and TGF-1β by Th2 cells in both two mice models. Histological examination of the pancreas revealed that EPE-treated mice had not only increased pancreatic insulin-expressing β cells (brown), and but also enhanced the percentage of Bcl-2 (green) / Bax (red) by IF staining analyses of islets compared with the S-NOD Con and the Cyp-NOD Con mice, implying that EPE displayed the protective effects of pancreas β cells. EPE-treated mice had increased the average immunoreactive system (IRS) score on insulin within the pancreas, and enhanced the numbers of the pancreatic islets. EPE displayed an improvement in the pancreas IRS scores and a decrease in proinflammatory cytokines. Moreover, EPE exerts blood glucose-lowering effects by regulating IL-17 expressions. Collectively, these results implied that EPE inhibits the development of autoimmune diabetes by regulating cytokine expression. Our results demonstrated that EPE had a therapeutic potential in the preventive effects of T1D and immunoregulation as a supplementary.
- As Reviewer suggests, “6. In abstract line number 27 authors write insulin was decreased (I think it is typo).”
And revised into
----> Our results showed that EPE-treated NOD mice or Cyp-accelerated NOD mice display a decrease in levels of blood glucose, insulin, and HbA1c, but an increase in blood insulin levels.
- As Reviewer suggests, “Animal ethical consideration part is missing.”
And revised into
---->
4.4. Animal Treatments
4.4.1. Part I. EPE or Vehicle Administered to S-NOD Mice
Fourteen female NOD/ShiLtJNarl mice (aged 3-week-old) were purchase from National Laboratory Animal Breeding Center and then housed in a pathogen-free environment. Animal study were approved and according to the guidelines of Central Taiwan University of Science and Technology, Taiwan, in accordance with the National Institutional Animal Care and Use Committee and approved by local animal ethics committee (Animal Ethics Committee, permit no. 109-CTUST-007). As our previous repot described [55], this regulation described the early euthanasia/ humane endpoints for animals those became severely ill to reduce pain to the minimum degree, the clinical signs employed to decide when to euthanize the animals as the following issues: 1. the reduced body weights by loss of 25% original body weights; 2. for rodents: the reduced food intake within 3 days below 50% of normal ingestions; 3. weak or dying status; 5. organs of animals with severe loss of functions and with clinical symptoms; 6. tumor; 7. Existed pain and it could not be controlled following treatment with analgesics [55].
- As Reviewer suggests, “Conclusion part could be bit shorten.”
And revised into
---->
Conclusions
Our results showed that EPE-treated NOD mice or Cyp-accelerated NOD mice displayed a decrease in the levels of blood glucose and HbA1c, but an increase in blood levels. ELISA analysis showed that EPE decreased the blood levels of IFN-γ and TNF-α by Th1 cells and reduced IL-1β and IL-6 by Th17 cells but increased IL-4, IL-10, and TGF-1β by Th2 cells in both two mice models. Western blotting analyses of the pancreatic target gene expressions showed that EPE inhibited the expression levels of inflammatory cytokines, including IFN-γ and TNF-α, by Th1 cells but upregulated the expression levels of IL-4, IL-10, and TGF-1β by Th2 cells in both two mice models. In both two mice models, histological examination of the pancreas revealed that EPE-treated mice had increased pancreatic insulin-expressing β cells (brown), the average IRS score of insulin, and the numbers of the pancreatic islets, and increased the percentage (%) of Bcl-2/Bax to avoid the destroy of pancreas β cells. Flow cytometric data showed that EPE-treated mice in both two mice had decreased the numbers of CD4+ IL-17 cells distribution per 10000 cells (%), but increased the numbers of CD4+ IL-4 and CD4+Foxp3+ by inhibition of Th1 cells but stimulation of Th2 cells, and regulation of Th17 cells convergently to secret proinflammatory cytokines (such as lowering blood levels of IL-1β and IL-6, and whereas increasing TGF-β1) and an impact on mutual restraint and interaction of Th1, Th2, and Th17 cell, thus contributing to the preventive effects of EPE on TID and immunoregulation.
In conclusion (Figure 11), these results implied that EPE inhibits the development of autoimmune diabetes by regulating cytokine expression. EPE displays an improvement in the pancreas immunoreactive scores and a decrease in proinflammatory cytokines. EPE exerts blood glucose-lowering effects by regulating IL-17 expressions. Our results demonstrated that EPE had a therapeutic potential in the preventive effects of T1D and immunoregulation.
- As Reviewer suggests, “In methods part it is not clear the age of mice when they first received the treatment.”
And revised into
----> (have described in the original manuscript:)
4.4.2. Part II. EPE or Vehicle-Administered to Cyp-NOD Mice
Twenty-four female NOD/ShiLtJNarl mice (aged 3-week-old) were purchase from National Laboratory Animal Breeding Center…. After one-week adaptation period, all mice were randomly divided
- As Reviewer suggests, “Figure 1 is not mentioned or referred in the text anywhere in the manuscript.“
And revised into
----> (have described in the original manuscript:)
Introduction
(1). Phylanthus emblica L. (Figure 1A) is widely distributed in…
(2). The preparation of ethyl acetate extract of P. emblica L. (EPE) was described as Figure 1B. In the present study, we employed both models of spontaneous non-obese diabetes (S-NOD) and cyclophosphamide-accelerated NOD (Cyp-NOD) mice to clarify the antidiabetic and immunoregulatory activities
Materials and Methods
4.2. Fruit Materials
Fruits of Phyllanthus emblica L. (Figure 1A) were purchased from Taiwan Miaoli County, Ogan marketing Cooperation. These fruits are identified by China Medical University, Taiwan, and the voucher specimen (CMPF385) is replaced.
4.3. Preparation of Ethyl Acetate Extract of Phyllanthus emblica L. (EPE)
The 7.3 kg fruits of Phyllanthus emblica L. (Figure 1A) were dried and pulverized into 1.34 kg fine powder. The fine powder was extracted with 6.3 L methanol three times at room temperature to concentration under vacuum, and then the crude methanolic extract (331.86 g) was obtained. The crude methanolic extract was subjected to three times of suspension in H2O and partition with EtOAc, respectively, and followed by concentration under reduced pressure, and then the H2O fraction (323.71 g) and the EtOAc fraction (47.28 g) was obtained (Figure 1B). The EtOAc fraction was employed for animal study.
- As Reviewer suggests, “Comments on the Quality of English Language
Language should be improved.“
And revised into
----> (The manuscript had already improved by American Journal of Editing (AJE) no. FFAA-4621-5E22-8DC8-F8B8) in Nov 15 2022) We could improve on the Quality of English Language by IJMS Journal assistants and I will pay, and nevertheless at the present because other Reviewer’s comments are too many points to improve this point in time, please give me additional time, thank you.

Round 2
Reviewer 2 Report
The authors made significant changes in the revised manuscript. Now it is in the acceptable form
Reviewer 3 Report
Authors have fixed all the concerned raised by me.
Minor changes required.